# Plant immune and growth receptors share common signalling components but localise to distinct plasma membrane nanodomains

Christoph A Bücherl[1], Iris K Jarsch[2†], Christian Schudoma[1‡], Cécile Segonzac[1§], Malick Mbengue[1¶], Silke Robatzek[1], Daniel MacLean[1], Thomas Ott[2**], Cyril Zipfel[1*]

[1]The Sainsbury Laboratory, Norwich Research Park, Norwich, United Kingdom; [2]Ludwig-Maximilians-Universität München, Institute of Genetics, Martinsried, Germany

*For correspondence: cyril.zipfel@tsl.ac.uk

Present address: †The Henry Wellcome Building of Cancer and Developmental Biology, Wellcome Trust/Cancer Research UK Gurdon Institute, University of Cambridge, Cambridge, United Kingdom; ‡Earlham Institute, Norwich, United Kingdom; §Plant Science Department, Plant Genome and Breeding Institute, Seoul National University, Seoul, Republic of Korea; ¶Laboratoire de Recherche en Sciences Végétales, Université de Toulouse, CNRS, UPS Auzeville, BP42617, 31326 Castanet Tolosan, Toulouse, France; **University of Freiburg, Faculty of Biology, Cell Biology, Freiburg, Germany

Competing interests: The authors declare that no competing interests exist.

**Abstract** Cell surface receptors govern a multitude of signalling pathways in multicellular organisms. In plants, prominent examples are the receptor kinases FLS2 and BRI1, which activate immunity and steroid-mediated growth, respectively. Intriguingly, despite inducing distinct signalling outputs, both receptors employ common downstream signalling components, which exist in plasma membrane (PM)-localised protein complexes. An important question is thus how these receptor complexes maintain signalling specificity. Live-cell imaging revealed that FLS2 and BRI1 form PM nanoclusters. Using single-particle tracking we could discriminate both cluster populations and we observed spatiotemporal separation between immune and growth signalling platforms. This finding was confirmed by visualising FLS2 and BRI1 within distinct PM nanodomains marked by specific remorin proteins and differential co-localisation with the cytoskeleton. Our results thus suggest that signalling specificity between these pathways may be explained by the spatial separation of FLS2 and BRI1 with their associated signalling components within dedicated PM nanodomains.

## Introduction

Multicellular organisms employ cell-surface receptors for surveying the environment and adjusting to changing physiological conditions. In plants, the repertoire of cell surface receptors has been considerably expanded and receptor kinases (RKs) form one of the largest protein families with over 600 members in *Arabidopsis thaliana* (hereafter, Arabidopsis) (*Shiu and Bleecker, 2001*). The schematic architecture of plant RKs is similar to that of animal receptor tyrosine kinases (RTKs); comprising an extracellular ligand binding domain, a single transmembrane helix, and an intracellular kinase domain (*Shiu and Bleecker, 2001*). Prominent examples of plant RKs are the immune receptor FLAGELLIN SENSING 2 (FLS2) (*Gómez-Gómez and Boller, 2000*) and the growth receptor BRASSINOSTEROID INSENSITIVE 1 (BRI1) (*Clouse et al., 1996*; *Li and Chory, 1997*). FLS2 is a pattern recognition receptor (PRR) that perceives the pathogen-associated molecular pattern (PAMP) flg22, an immunogenic epitope of bacterial flagellin, to initiate PAMP-triggered immunity (PTI) (*Felix et al., 1999*; *Zipfel et al., 2004*; *Chinchilla et al., 2006*; *Boller and Felix, 2009*). BRI1 binds brassinosteroids (BRs), a class of phytohormones involved in various aspects of plant growth and development (*Kinoshita et al., 2005*; *Kim and Wang, 2010*; *Singh and Savaldi-Goldstein, 2015*).

**eLife digest** Unlike most animals, plants cannot move away if their environment changes for the worse. Instead, a plant must sense these changes and respond appropriately, for example by changing how much it grows. Disease-causing microbes in the immediate environment represent another potential threat to plants. To detect these microbes, plant cells have proteins called "pattern recognition receptors" in their surface membranes that sense certain molecules from the microbes (similar receptors are found in animals too). When a receptor protein recognises one such microbial molecule, it becomes activated and forms a complex with other proteins referred to as co-receptors. The protein complex then sends a signal into the cell to trigger an immune response.

Plants also use similar receptor proteins to sense their own signalling molecules and regulate their growth and development. These growth-related receptors rely on many of the same co-receptors and signalling components as the immunity-related receptors. This posed the question: how can plant cells use the same proteins to trigger different responses to different signals?

Bücherl et al. have now used high-resolution microscopy and the model plant *Arabidopsis thaliana* to show that the plant's immune receptors and growth receptors are found in separate clusters at the plant cell's surface membrane. These clusters are only a few hundred nanometres wide, and they also contained other signalling components that are needed to quickly relay the signals into the plant cell.

Bücherl et al. suggest that, by organizing their receptors into these physically distinct clusters, plant cells can use similar proteins to sense different signals and respond in then different ways. This idea will need to be tested in future studies. Further work is also needed to understand how these clusters of signalling proteins are assembled and inserted at specific locations within the surface membrane of a plant cell.

Despite their different biological functions, FLS2- and BRI1-mediated signalling pathways share several similarities, in particular at or close to the plasma membrane (PM). The PM is the cellular compartment, where both receptors localise to (*Robatzek et al., 2006*; *Friedrichsen et al., 2000*), where they bind their respective ligands flg22 or BRs (*Gómez-Gómez et al., 2001*; *Bauer et al., 2001*; *Kinoshita et al., 2005*), and where presumably their main signalling activity is executed (*Smith et al., 2014*; *Irani et al., 2012*). Although FLS2 and BRI1 are competent for ligand binding via their extracellular leucine-rich repeat (LRR) domains, they rely on SOMATIC EMBRYOGENESIS RECEPTOR-LIKE KINASE (SERK) co-receptors for signalling initiation (*Nam and Li, 2002*; *Li et al., 2002*; *Chinchilla et al., 2007*; *Heese et al., 2007*; *Roux et al., 2011*; *Gou et al., 2012*), which are also LRR-RKs (*Aan den Toorn et al., 2015*). Structural and biochemical analysis of FLS2- and BRI1-SERK hetero-oligomers revealed that flg22 and BRs act as 'molecular glues' that stabilise or induce receptor complexes (*Sun et al., 2013*; *She et al., 2011*; *Hothorn et al., 2011*). Ligand binding additionally triggers auto- and trans-phosphorylation events within the receptor complexes (*Schulze et al., 2010*; *Wang et al., 2008*) and, in the case of BRI1, also the release of inhibitory mechanisms (*Wang and Chory, 2006*; *Jaillais et al., 2011*). After gaining their full kinase activities, FLS2 and BRI1 receptor complexes initiate phosphorylation cascades that culminate in flg22- or BR-responsive transcriptional regulation (*Guo et al., 2013*; *Li et al., 2016*). The relay of phosphorylation signals from the PM to the nucleus involves receptor-like cytoplasmic kinases (RLCKs) that can associate to the PM and that are direct substrates of the ligand-binding receptor complexes (*Lin et al., 2013*; *Belkhadir and Jaillais, 2015*; *Couto and Zipfel, 2016*). Similar to the SERK co-receptors, the RLCKs BRASSINOSTEROID SIGNALING KINASE 1 (BSK1) and BOTRYTIS-INDUCED KINASE 1 (BIK1) are common signalling components in both pathways. Whereas BSK1 is a positive regulator for both signalling routes (*Tang et al., 2008*; *Shi et al., 2013*), BIK1 is a positive regulator for PTI responses (*Lu et al., 2010*; *Zhang et al., 2010*), but a negative regulator for BR signalling (*Lin et al., 2013*).

Even though FLS2- and BRI1-mediated signalling pathways have been extensively studied genetically and biochemically, little is known about how FLS2 and BRI1 are organised within the PM and how these two receptors fulfil their sensory activity at the cell periphery. In contrast to the original fluid mosaic model (*Singer and Nicolson, 1972*), which considered the PM as a two-dimensional

liquid composed of a lipid bilayer that is interspersed by integral or associated proteins, it is nowadays accepted that the PM is a highly structured and dynamic cellular compartment organised at three hierarchic levels (*Kusumi et al., 2011*; *Nicolson, 2014*). The first level of PM organisation is characterised by the interaction of the lipid bilayer with the underlying cortical cytoskeleton, the second level by protein-lipid interactions with the PM, and the third level of PM organisation is the result of protein-protein interactions that lead to formation of PM-associated or -integral homo- and hetero-oligomers (*Kusumi et al., 2011*), *e.g.* FLS2- or BRI1-SERK3/BAK1 complexes. In plants, the cell wall has additional influence on the PM organisation and dynamics (*Martinière et al., 2012*). As a consequence, lateral mobility and distribution of lipids and proteins within the PM is highly heterogeneous leading to the formation of dynamic protein clusters and PM sub-compartments with different shapes and sizes (*Jaqaman and Grinstein, 2012*; *Jarsch et al., 2014*; *Ziomkiewicz et al., 2015*). Each compartment or domain provides specific biophysical and biochemical environments for its residents and thus directly influences associated signalling activities (*Kusumi et al., 2012*; *Saka et al., 2014*; *Garcia-Parajo et al., 2014*; *Tapken and Murphy, 2015*).

With regard to PM signalling, specialised PM areas often referred to as PM nanodomains have attracted particular attention. For example, EPIDERMAL GROWTH FACTOR RECEPTOR (EGFR), a mammalian RTK, forms clusters that co-localise with PM nanodomains, and EGFR cluster formation depends on the integrity of the PM lipid composition (*Gao et al., 2015*). A recent report showed that BRI1 also localises to PM nano- or micro-domains in Arabidopsis roots and that partitioning of BRI1 into different PM microdomains is crucial for BR signalling (*Wang et al., 2015*). Due to the numerous similarities between BRI1 and FLS2 signalling initiation, we were interested in elucidating the localisation patterns of these two LRR-RKs in a comparative fashion within the highly structured PM of plant cells.

Here, we investigated the steady-state PM organisation of FLS2 and BRI1 receptors using live-cell imaging, single-particle tracking and quantitative co-localisation analysis in the PM of leaf epidermal cells, a cell file, in which both receptors are similarly expressed. Our results showed that FLS2 and BRI1 are heterogeneously distributed and that both receptors form transient receptor clusters. Although the spatial characteristics of FLS2 and BRI1 receptor clusters were similar, we observed differences in their dynamic behaviour with FLS2 clusters being more stable. Moreover, we detected only a limited overlap between the two receptor populations. This finding was confirmed by visualising FLS2 and BRI1 clusters within distinct remorin-labelled PM nanodomains. We additionally investigated the PM localisation patterns of BSK1 and BIK1 in complex with the two ligand-binding LRR-RKs. Imaging of FLS2 and BRI1 signalling complexes revealed a confined PM localisation and cluster formation. Importantly, BIK1 signalling complexes localised differentially, whereby BRI1-BIK1, but not FLS2-BIK1, complexes associated with cortical microtubules. Together, our data suggest that the distinct spatiotemporal localisation of FLS2 and BRI1 within specialised PM nanodomains may contribute to signalling specificity between immune and growth signalling mediated by these receptors.

## Results

### FLS2 and BRI1 form receptor clusters within the PM

To obtain an overview of the PM distribution of FLS2 and BRI1, we investigated two stable transgenic Arabidopsis lines that express the C-terminally GFP-tagged receptors under their native promoters (*Göhre et al., 2008*; *Geldner et al., 2007*). Live-cell imaging using confocal laser scanning microscopy (CLSM) revealed that FLS2 and BRI1 were heterogeneously distributed within the PM (*Figure 1A and B*). Both receptors formed dispersed punctate structures with increased fluorescence intensities. This local concentration of FLS2 and BRI1 subpopulations within the PM indicates the formation of receptor clusters, a phenomenon known from mammalian transmembrane receptors or helper proteins, such as EGFR (*Clayton et al., 2007*) or LINKER OF ACTIVATED T CELLS (LAT) (*Su et al., 2016*). To emphasis our observation of receptor clusters, we processed the presented images using a spot-enhancing filter described by *Sage et al. (2005)* and applied the 'fire' lookup table (*Figure 1—figure supplement 1*). Importantly, we observed similar cluster formation of FLS2 and BRI1 whether upon native expression in Arabidopsis or after transient expression in *Nicotiana benthamiana* (*N. benthamiana*) (*Figure 1C and D*). This feature enabled us to perform further co-localisation analysis in this heterologous system.

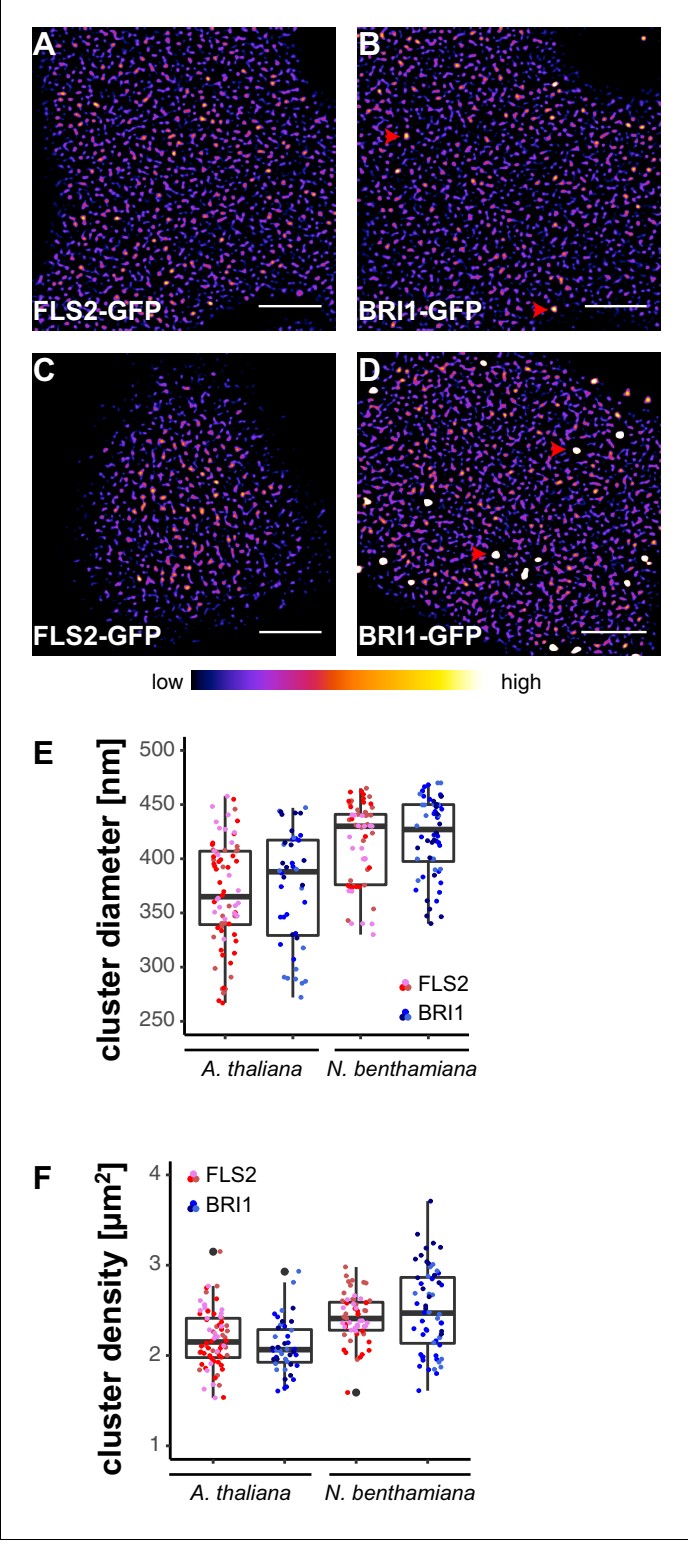

**Figure 1.** FLS2 and BRI1 form receptor clusters within the plasma membrane. (A, B) Plasma membrane localisation of FLS2-GFP (A) and BRI1-GFP (B) in epidermal cells of Arabidopsis seedling cotyledons. (C, D) Plasma membrane localisation of FLS2-GFP (C) and BRI1-GFP (D) after transient expression in epidermal leaf cells of *N. benthamiana*. (E) Quantification of FLS2-GFP and BRI1-GFP plasma membrane receptor cluster diameters in epidermal cells of Arabidopsis cotyledons and after transient expression in *N. benthamiana* leaves. The coloured data points represent the technical replicates of 3 independent experiments. No statistical differences were observed based

*Figure 1 continued on next page*

*Figure 1 continued*

on two-tailed heteroscedastic *t*-tests and a Bonferroni multiple hypothesis correction. (**F**) Quantification of FLS2-GFP and BRI1-GFP plasma membrane receptor cluster densities in epidermal cells of Arabidopsis seedlings and after heterologous expression in *N. benthamiana*. The coloured data points represent the technical replicates of 3 independent experiments. No statistical differences were observed based on two-tailed heteroscedastic *t*-tests and a Bonferroni multiple hypothesis correction. The presented images were acquired using confocal laser scanning microscopy (CLSM). Scale bars represent 5 μm. The colour bar represents the colour code for fluorescence intensities. Black dots represent outliers. Red arrowheads indicate endosomal compartments of BRI1-GFP. Endosomal compartments were omitted for quantitative analysis.

The following figure supplements are available for figure 1:

**Figure supplement 1.** Illustration of the image processing steps for emphasising receptor cluster formation.

**Figure supplement 2.** Plasma membrane localisation of FLS2 and BRI1 with respect to the cytoskeleton.

Besides residing within the PM, BRI1-GFP additionally localised to mobile endomembrane compartments with high fluorescence intensities (highlighted by red arrowheads in *Figure 1B and D*). These intracellular compartments most likely resembled trans-Golgi network/early endosomes and late endosomes/multi-vesicular bodies (*Geldner et al., 2007*; *Irani et al., 2012*).

In contrast to the highly mobile endomembrane compartments, the visualised FLS2- and BRI1-GFP clusters appeared rather immobile and seemed to follow a certain spatial organisation. FLS2 and BRI1 clusters often aligned in pearl chain-like structures and interconnected lines with low fluorescence intensities frequently separated by PM areas containing several receptor clusters, in particular for FLS2-GFP (*Figure 1—figure supplement 2*).

It is known that the cytoskeleton influences the hierarchic PM organisation (*Plowman et al., 2005*; *Kusumi et al., 2011*) and therefore we visualised FLS2 and BRI1 in the presence of actin and microtubule markers. As shown in *Figure 1—figure supplement 2*, pearl chain-like aligned FLS2 and BRI1 clusters often coincided with the underlying actin cytoskeleton, whereas the interconnected lines with low fluorescence intensities overlapped with cortical microtubule filaments, both in Arabidopsis and *N. benthamiana*.

Next, we quantified the size and density of individual receptor clusters. The mean values (± standard deviation) for FLS2 receptor cluster diameters were 356 ± 49 nm and 387 ± 55 nm in Arabidopsis and *N. benthamiana*, respectively, compared to 372 ± 38 nm in Arabidopsis and 392 ± 37 nm in *N. benthamiana* for BRI1 (*Figure 1E* and *Supplementary file 1*). Although there was a slight trend towards increased receptor cluster sizes in *N. benthamiana*, no statistical difference was observed. On average, we observed 2.21 ± 0.33 FLS2 clusters in Arabidopsis and 2.23 ± 0.26 in *N. benthamiana* per μm$^2$, whereas the densities of BRI1 clusters were 2.02 ± 0.27 and 2.43 ± 0.39 per μm$^2$ in Arabidopsis and *N. benthamiana*, respectively (*Figure 1F* and *Supplementary file 1*). Thus, based on our confocal micrograph analysis, FLS2 and BRI1 showed similar spatial features. Both LRR-RKs were heterogeneously distributed within the PM and formed receptor clusters of comparable size and density. Only a difference with regard to the exclusion of receptor clusters from PM areas by cortical microtubules was observed, with FLS2 being more strongly affected.

In addition to confocal microscopy, we applied variable angle epifluorescence microscopy (VAEM) to further characterise the dynamic behaviour of PM-localised FLS2 and BRI1 receptors. VAEM is a technique related to total internal reflection microscopy (TIRF) (*Vizcay-Barrena et al., 2011*). In comparison to CLSM, VAEM has the advantages of improved z-resolution and fast image acquisition (*Vizcay-Barrena et al., 2011*). Therefore VAEM is ideal for studying protein localisation and protein dynamics within or close to the PM (*Wan et al., 2011*; *Vizcay-Barrena et al., 2011*). As shown in *Figure 2A–D*, single frame VAEM images resembled our observations made using CLSM (*Figure 1A–D*). We repeatedly observed dispersed FLS2- and BRI1-GFP clusters in the PM of Arabidopsis or *N. benthamiana* leaf epidermal cells. However, using VAEM we were able to acquire image time series (*Video 1* and *2*). Micrograph analysis using kymograph representations revealed increased stability and reduced mobility for FLS2-GFP compared to BRI1-GFP clusters, both in Arabidopsis and *N. benthamiana* (*Figure 2E*). These observations were confirmed using single-particle

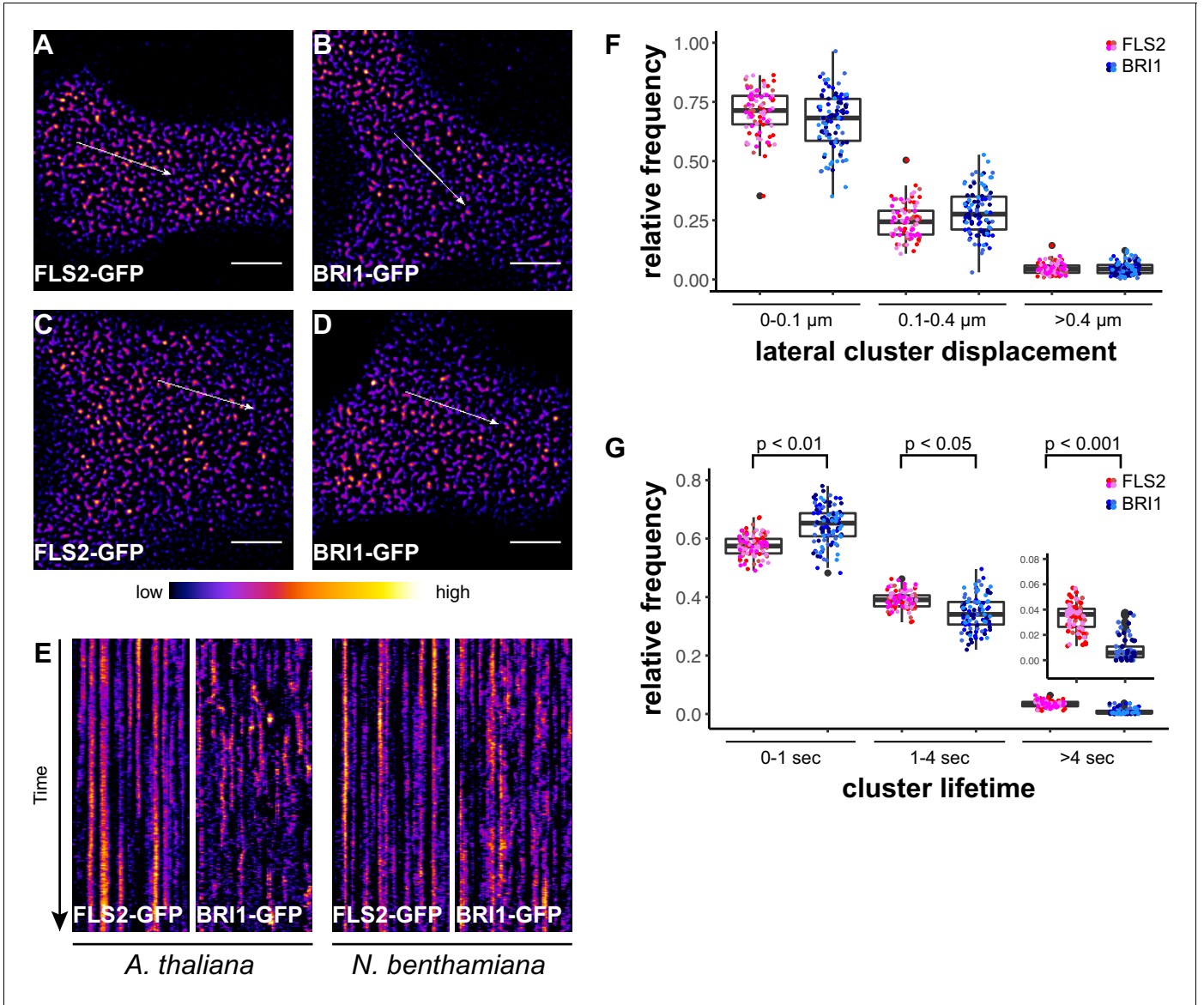

**Figure 2.** FLS2 receptor clusters are more stable than BRI1 clusters. (A, B) Plasma membrane localisation of FLS2-GFP (A) and BRI-GFP (B) in epidermal cells of Arabidopsis seedling cotyledons. (C, D) Plasma membrane localisation of FLS2-GFP (C) and BRI1-GFP (D) after transient expression in epidermal leaf cells of *N. benthamiana*. (E) Kymograph analysis of FLS2-GFP and BRI1-GFP plasma membrane receptor clusters in epidermal cells of Arabidopsis seedling cotyledons and after transient expression in *N. benthamiana*. Kymographs were obtained from VAEM time series with a temporal resolution of 0.5 s over 250 frames along the indicated arrows in micrographs (A) to (D). (F) Quantification of FLS2-GFP and BRI1-GFP receptor cluster displacements in epidermal cells of Arabidopsis seedling cotyledons obtained from VAEM time series with a temporal resolution of 0.5 s over 250 frames. The coloured data points represent the technical replicates of 4 independent experiments. No statistical differences were observed based on two-tailed heteroscedastic *t*-tests and a Bonferroni multiple hypothesis correction. (G) Quantification of FLS2-GFP and BRI1-GFP receptor cluster lifetimes in epidermal cells of Arabidopsis seedling cotyledons obtained from VAEM time series with a temporal resolution of 0.5 s over 250 frames. The coloured data points represent the technical replicates of 4 independent experiments. The indicated p-values were obtained using a two-tailed heteroscedastic *t*-test and a Bonferroni multiple hypothesis correction. The presented images were acquired using variable angle epi-fluorescence microscopy (VAEM). Scale bars represent 5 µm. The colour bar represents the colour code for fluorescence intensities. Black dots represent outliers.

The following figure supplement is available for figure 2:

**Figure supplement 1.** The stability of FLS2 clusters depends on the PM lipid composition.

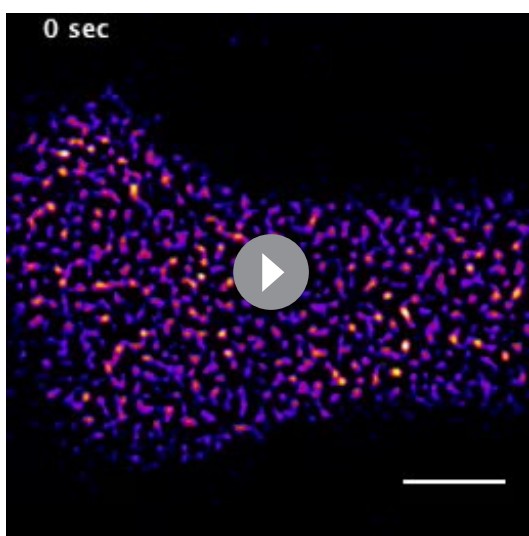

**Video 1.** Dynamics of FLS2 receptor clusters within the plasma membrane. The presented time series was acquired from an epidermal cell of an Arabidopsis seedling cotyledon expressing FLS2-GFP under its native promoter using variable angle epi-fluorescence microscopy (VAEM). The acquisition time was 0.5 s per frame over 250 frames in total. The scale bar represents 5 µm.

First, we deprived seedlings from endogenous BRs by cultivating them for 2 days in liquid medium containing 5 µM brassinazole (BRZ), an inhibitor of brassinosteroid biosynthesis (*Asami et al., 2000*). As shown in *Figure 3*, depletion of endogenous BRs resulted in increased BRI1 cluster mobility (*Figure 3A*). In contrast, the lateral displacement of FLS2 clusters was not affected (*Figure 3A*). The determined cluster lifetimes for FLS2 and BRI1 showed a similar distribution as under non-treated conditions with a slight trend towards increased stability for BRI1 clusters upon BZR treatment (*Figure 3B*). Subsequently, we activated BRI1-mediated signalling by exogenous application of 100 nM 24-epi-brassinolide (BL). In analogy to BRZ-treatment, BL had no effect on FLS2 clusters but decreased the lateral displacement of BRI1 receptor clusters within a time frame of 30 min (*Figure 3C–H*).

Accordingly, we observed a reduction in lateral cluster displacement for FLS2 receptors after seedlings were exposed to 100 nM flg22, whereas BRI1 was unaffected (*Figure 4*). Again, there was no quantitative effect on the receptor cluster lifetimes of both receptors (*Figure 4*).

Our findings that receptor activation leads to reduced lateral mobility within the PM are

tracking. BRI1-GFP showed a trend towards increased lateral cluster displacement (*Figure 2F*) and FLS2-GFP showed an increased population of long-lived receptor clusters (*Figure 2G*). Additionally, we noticed that some FLS2 and BRI1 clusters suddenly appeared or disappeared from the PM. These observations most likely reflect constitutive exo- and endocytosis processes (*Beck et al., 2012*; *Martins et al., 2015*; *Wang et al., 2015*).

Taken together, we observed that FLS2 and BRI1 formed non-randomly distributed receptor clusters across the PM. These receptor clusters were comparable in size and density; however, showed differences with regard to dynamic features.

## Receptor activation reduces the lateral receptor cluster mobility

Using our non-treated conditions, we visualised both inactive FLS2 populations and partially active BRI1 receptors given the presence of endogenous BRs. This experimental setup most likely best reflects the steady-state configuration of both receptors in a natural situation prior to pathogen attack. Still, we were also interested to test how ligand availability influences the dynamic behaviour of FLS2 and BRI1 receptor clusters.

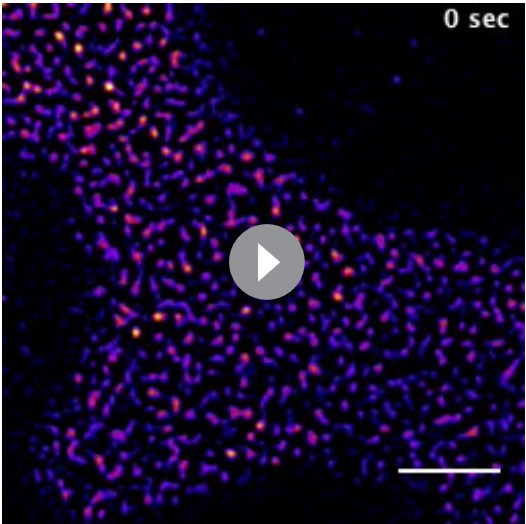

**Video 2.** Dynamics of BRI1 receptor clusters within the plasma membrane. The presented time series was acquired from an epidermal cell of an Arabidopsis seedling cotyledon expressing BRI1-GFP under its native promoter using variable angle epi-fluorescence microscopy (VAEM). The acquisition time was 0.5 s per frame over 250 frames in total. The scale bar represents 5 µm.

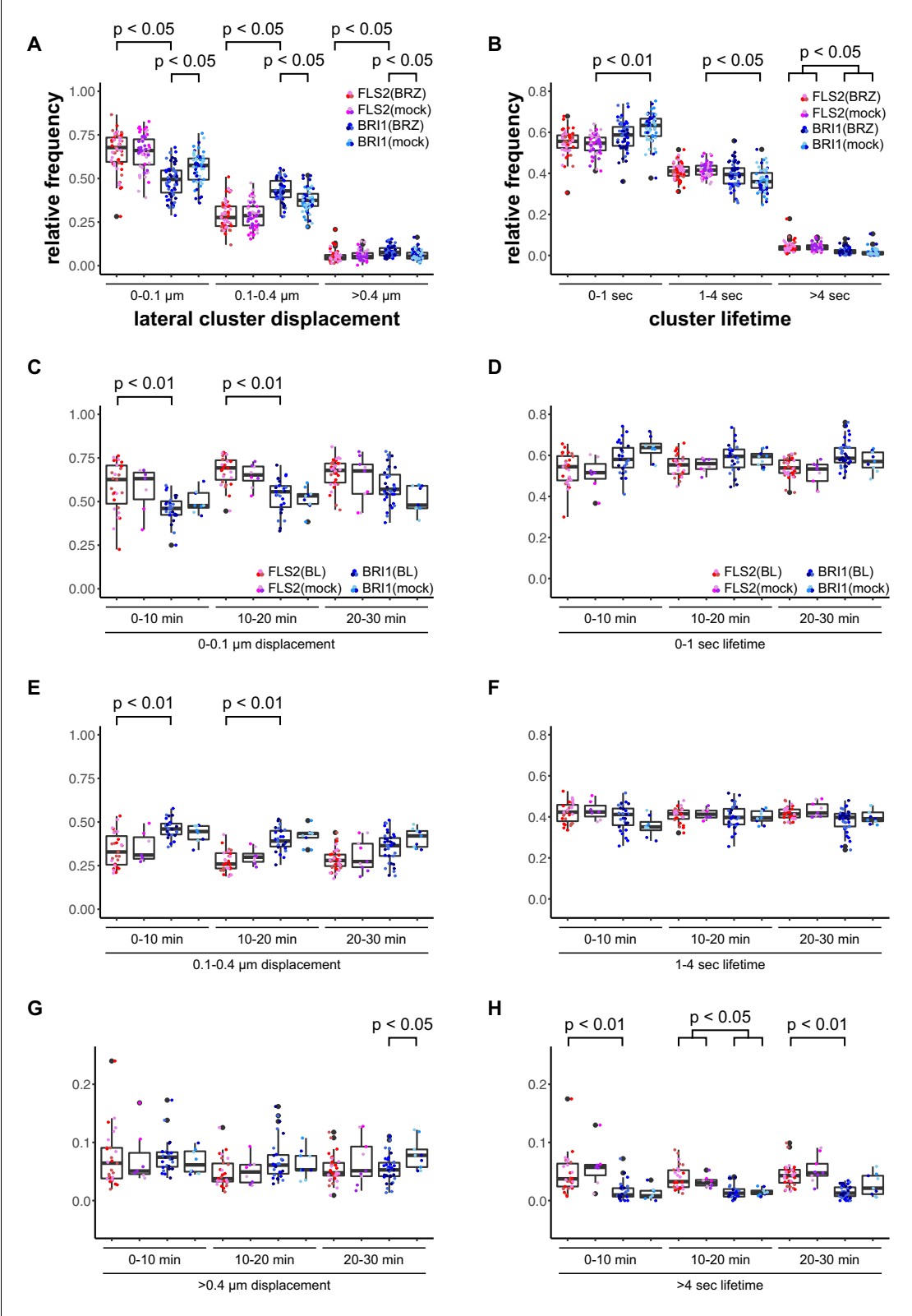

**Figure 3.** BRs reduce BRI1 cluster displacement within the plasma membrane. (**A**) Quantification of FLS2-GFP and BRI1-GFP receptor cluster displacements in epidermal cells of Arabidopsis seedling cotyledons after 2 days in liquid medium containing 5 μM BRZ. (**B**) Quantification of FLS2-GFP and BRI1-GFP receptor cluster lifetimes in epidermal cells of Arabidopsis seedling cotyledons after 2 days in liquid medium containing 5 μM BRZ. (**C**) Time-dependent quantification of short-range FLS2-GFP and BRI1-GFP receptor cluster displacements in epidermal cells of Arabidopsis seedling

*Figure 3 continued on next page*

*Figure 3 continued*

cotyledons after BRZ-treatment and subsequent application of 100 nM BL. (D) Time-dependent quantification of short FLS2-GFP and BRI1-GFP receptor cluster lifetimes in epidermal cells of Arabidopsis seedling cotyledons after BRZ-treatment and subsequent application of 100 nM BL. (E) Time-dependent quantification of medium-range FLS2-GFP and BRI1-GFP receptor cluster displacements in epidermal cells of Arabidopsis seedling cotyledons after BRZ-treatment and subsequent application of 100 nM BL. (F) Time-dependent quantification of medium FLS2-GFP and BRI1-GFP receptor cluster lifetimes in epidermal cells of Arabidopsis seedling cotyledons after BRZ-treatment and subsequent application of 100 nM BL. (G) Time-dependent quantification of long-range FLS2-GFP and BRI1-GFP receptor cluster displacements in epidermal cells of Arabidopsis seedling cotyledons after BRZ-treatment and subsequent application of 100 nM BL. (H) Time-dependent quantification of long FLS2-GFP and BRI1-GFP receptor cluster lifetimes in epidermal cells of Arabidopsis seedling cotyledons after BRZ-treatment and subsequent application of 100 nM BL. The presented data points were obtained from VAEM time series with a temporal resolution of 0.5 s over 250 frames. The coloured data points represent the technical replicates of 3 independent experiments. The indicated p-values were obtained using a one-tailed heteroscedastic *t*-test and a Bonferroni multiple hypothesis correction.

contrary to *Wang et al. (2015)*, who reported increased BRI1 diffusion after ligand application in Arabidopsis roots, but in line with observations made for the plant receptors FLS2 and LYK3 (*Ali et al., 2007*; *Haney et al., 2011*) as well as the mammalian cell surface receptor EGFR (*Low-Nam et al., 2011*). Thus, FLS2 and BRI1 exhibited comparable behaviour in response to their respective ligands, nonetheless, the dynamic features of both cluster populations still differed from each other.

## FLS2 and BRI1 show distinct plasma membrane localisation patterns

To address whether FLS2 and BRI1 clusters coincide or are spatially separated within the PM, we performed co-localisation studies. As positive control, we first determined the overlap of two differently tagged FLS2 receptor populations. We co-expressed FLS2-GFP and FLS2-mCherry (*Mbengue et al., 2016*) in leaf epidermal cells of *N. benthamiana* and, as shown in *Figure 5A–D*, both fluorescently tagged FLS2 populations showed similar PM localisation patterns and also co-localised (*Figure 5C and D*). Based on quantitative co-localisation analysis, we determined moderate to high Pearson correlation coefficients for FLS2-GFP and FLS2-mCherry fluorescence signals (*Figure 5I* and *Figure 5—figure supplement 1*).

Subsequently, we compared the distributions of BRI1-GFP and FLS2-mCherry receptors (*Figure 5E–H*). Quantitative image analysis of the obtained image series indicated a strongly reduced co-localisation between BRI1-GFP and FLS2-mCherry when compared to the FLS2-GFP/FLS2-mCherry combination (*Figure 5A–D*). As shown in *Figure 5I*, we determined Pearson correlation coefficients of around zero, which represents non-correlated localisation or no co-localisation (*McDonald and Dunn, 2013*). Based on our previous findings, which indicated increased lateral BRI1 mobility by depletion of endogenous BRs, we also analysed the co-localisation of both receptors after BRZ-treatment. However, we observed the same co-localisation pattern as under non-treated steady-state conditions (*Figure 5J*), suggesting a ligand-independent spatial separation of FLS2 and BRI1 receptors. These results were confirmed using image randomisation and by using BRI1-mRFP as reference (*Figure 5—figure supplement 1* and *Supplementary file 1*). Consequently, our quantitative co-localisation analysis revealed distinct immune and growth receptor clusters within the PM of leaf epidermal cells.

To obtain a more dynamic view on the co-localisation or spatial separation between the FLS2 and BRI1 receptor populations, we additionally applied dual-colour VAEM on leaf epidermal cells that co-expressed BRI1-GFP and FLS2-mCherry (*Video 3* and *4*, *Figure 6*). We hardly observed overlap between the two LRR-RKs as indicated by the kymograph representation in *Figure 6H*.

Collectively, our co-localisation analysis revealed that the vast majority of the two LRR-RKs formed distinct receptor clusters that were spatiotemporally separated.

## FLS2 and BRI1 co-localise differentially with remorin nanodomain markers

So far, we provided evidence for the formation of distinct FLS2 and BRI1 clusters that were heterogeneously distributed within the PM of leaf epidermal cells. Considering the proposed hierarchic organisation of the PM (*Kusumi et al., 2011*), one could assume that these receptor clusters may

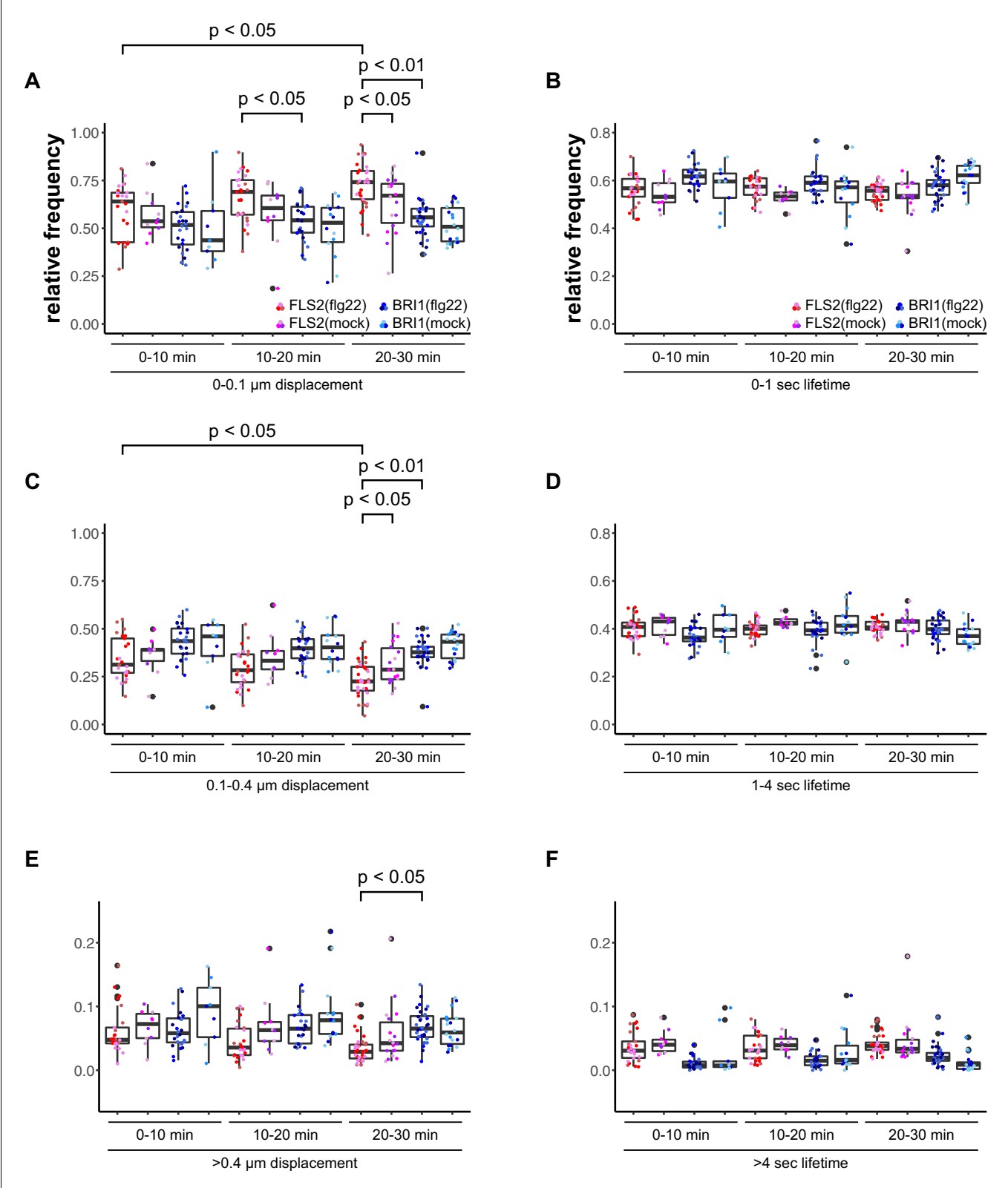

**Figure 4.** Activation of FLS2 results in reduced lateral receptor cluster displacement. (**A**) Time-dependent quantification of short-range FLS2-GFP and BRI1-GFP receptor cluster displacements in epidermal cells of Arabidopsis seedling cotyledons after application of 100 nM flg22. (**B**) Time-dependent quantification of short FLS2-GFP and BRI1-GFP receptor cluster lifetimes in epidermal cells of Arabidopsis seedling cotyledons after application of 100 nM flg22. (**C**) Time-dependent quantification of medium-range FLS2-GFP and BRI1-GFP receptor cluster displacements in epidermal cells of

*Figure 4 continued on next page*

*Figure 4 continued*

Arabidopsis seedling cotyledons after application of 100 nM flg22. (**D**) Time-dependent quantification of medium FLS2-GFP and BRI1-GFP receptor cluster lifetimes in epidermal cells of Arabidopsis seedling cotyledons after application of 100 nM flg22. (**E**) Time-dependent quantification of long-range FLS2-GFP and BRI1-GFP receptor cluster displacements in epidermal cells of Arabidopsis seedling cotyledons after application of 100 nM flg22. (**F**) Time-dependent quantification of long FLS2-GFP and BRI1-GFP receptor cluster lifetimes in epidermal cells of Arabidopsis seedling cotyledons after application of 100 nM flg22. The presented data points were obtained from VAEM time series with a temporal resolution of 0.5 s over 250 frames. The coloured data points represent the technical replicates of 3 independent experiments. The indicated p-values were obtained using a one-tailed heteroscedastic *t*-test and a Bonferroni multiple hypothesis correction.

reside within PM nanodomains. In fact, this was recently shown for BRI1 in root epidermal cells by co-localising it with the PM nanodomain marker AtFLOT1 (*Wang et al., 2015*). Interestingly, proteomic studies identified two remorin (REM) proteins, REM1.2 and REM1.3, that were enriched in detergent-resistant membranes upon flg22 application (*Keinath et al., 2010*). REMs form a large plant-specific protein family with characteristic PM nanodomain localisation patterns (*Jarsch et al., 2014*). Moreover, different members of this family have previously been associated with plant-microbe interactions (*Raffaele et al., 2009*; *Lefebvre et al., 2010*; *Tóth et al., 2012*; *Perraki et al., 2012, 2014*; *Bozkurt et al., 2014*) and with SERK-dependent processes (*Gui et al., 2016*).

Based on the findings of *Keinath et al. (2010)*, we investigated the spatial relationship between the two LRR-RKs and REM1.2 in leaf epidermal cells. As shown in *Figure 7A I–A III and I*, we identified a positive correlation for FLS2-GFP and mRFP-REM1.2 fluorescence intensities. In contrast, the co-localisation of this PM nanodomain marker with BRI1 was unspecific (*Figure 7B I−B III and I*, *Figure 7—figure supplement 1*). Similar results were obtained using mRFP-REM1.3 as PM nanodomain marker (*Figure 7C I–D III and I*, *Figure 7—figure supplement 1*).

Besides REM1.2 and REM1.3, *Benshop et al. (2007)* identified REM6.1 and REM6.2 as flg22-induced phosphoproteins. Therefore, we investigated also the co-localisation of FLS2 and BRI1 with these two REM proteins. The results are shown in *Figure 7E I* to *Figure 7H III*. Similar to REM1.2 and REM1.3, we revealed a positive correlation between FLS2-GFP and mRFP-REM6.1 protein populations. However, REM6.1-labelled PM nanodomains also harboured a considerable amount of BRI1 receptors (*Figure 7F III and I*, *Figure 7—figure supplement 1*). For mRFP-REM6.2, we observed unexpectedly the opposite behaviour (*Figure 7G I–H III*). REM6.2-positive PM nanodomains contained a significantly elevated amount of BRI1 compared to FLS2 receptors. The results of quantitative co-localisation analysis are summarised in *Figure 7I*.

Taken together, these findings demonstrated that the heterogeneously distributed FLS2 and BRI1 receptor clusters are indeed residing within PM nanodomains of leaf epidermal cells. The differential co-localisation of the two LRR-RKs with regard to the tested REM marker proteins further emphasised the spatial separation and distinct localisation of FLS2 and BRI1 receptors.

## BRI1-BIK1, but not FLS2-BIK1, complexes associate with cortical microtubules

Our cell biological study indicated a spatial separation between immune and growth receptors in steady-state conditions. Though, genetically and biochemically there exist apparent connections between FLS2- and BRI1-mediated signalling pathways, although many of these interconnections seem to occur at the transcriptional level (*Albrecht et al., 2012*; *Belkhadir et al., 2012, 2014*; *Lozano-Durán et al., 2013*; *Fan et al., 2014*; *Malinovsky et al., 2014*; *Lozano-Durán and Zipfel, 2015*; *Jiménez-Góngora et al., 2015*). Nevertheless, both receptors depend on additional PM-localised or PM-associated signalling components for relaying the information of ligand binding to the extracellular LRRs domains across the PM and into the cell interior. Therefore we assume that a signalling competent unit contains at least one ligand-binding receptor, one (or several) co-receptor(s), and one (or several) RLCK(s). Intriguingly, FLS2 and BRI1 employ, at least from a genetic perspective, the same components; SERK co-receptors (*Nam and Li, 2002*; *Li et al., 2002*; *Chinchilla et al., 2007*; *Heese et al., 2007*; *Roux et al., 2011*; *Gou et al., 2012*), and the RLCKs BSK1 and BIK1 (*Tang et al., 2008*; *Shi et al., 2013*; *Lu et al., 2010*; *Zhang et al., 2010*; *Lin et al., 2013*).

To investigate the spatial organisation of FLS2 and BRI1 signalling units, we made use of bimolecular fluorescence complementation (BiFC). Since epitope-tagging of BAK1/SERK3 (and potentially

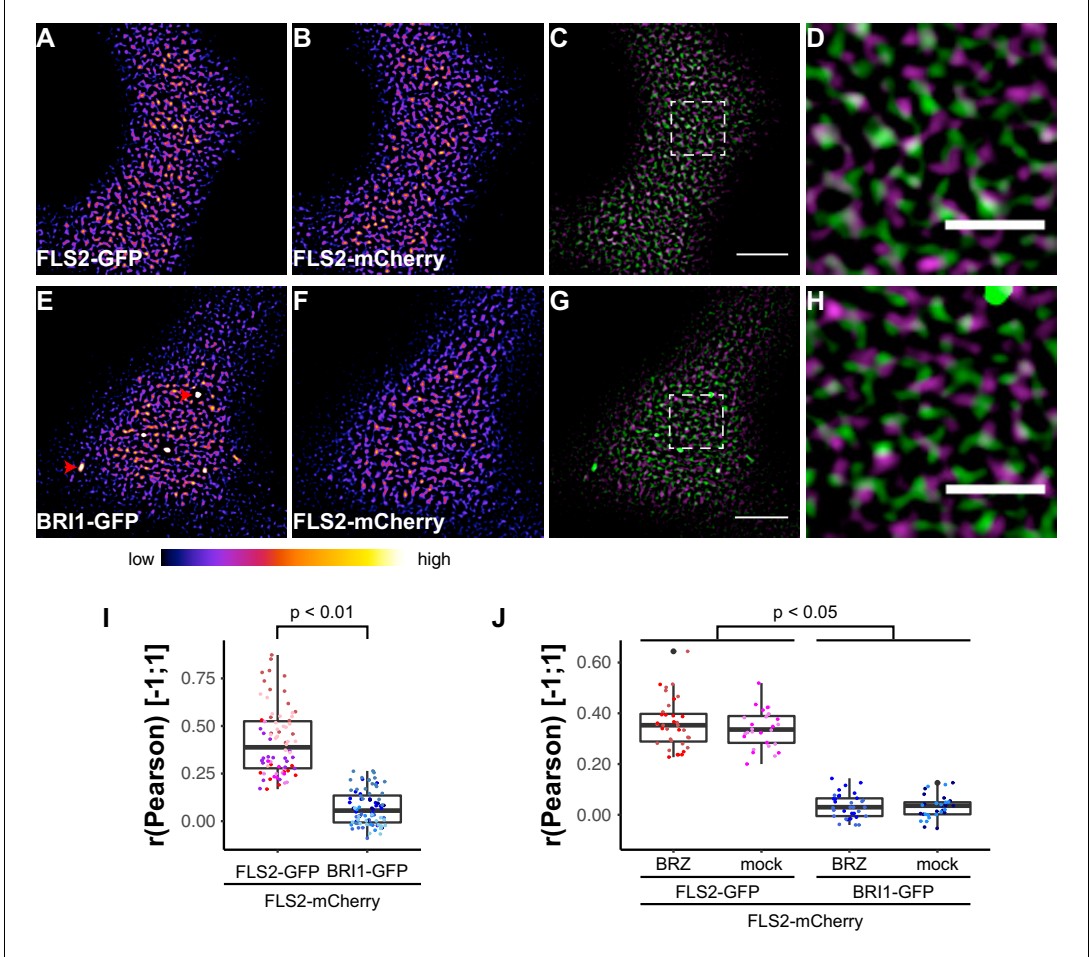

**Figure 5.** FLS2 and BRI1 show distinct plasma membrane localisation patterns. (**A–D**) Confocal micrographs of FLS2-GFP (**A**) and FLS2-mCherry (**B**) plasma membrane localisation after transient co-expression in epidermal leaf cells of *N. benthamiana* as well as the merged image (**C**) and an image inset (**D**). (**E–H**) Confocal micrographs of BRI1-GFP (**E**) and FLS2-mCherry (**F**) plasma membrane localisation after transient co-expression in epidermal leaf cells of *N. benthamiana* as well as the merged image (**G**) and an image inset (**H**). (**I**) Quantitative co-localisation analysis for FLS2-GFP or BRI1-GFP, respectively, with FLS2-mCherry after transient co-expression in epidermal leaf cells of *N. benthamiana.* The coloured data points represent the technical replicates of 6 independent experiments. The indicated p-values were obtained using a two-tailed heteroscedastic *t*-test and a Bonferroni multiple hypothesis correction. (**J**) Quantitative co-localisation analysis for FLS2-GFP or BRI1-GFP, respectively, with FLS2-mCherry after transient co-expression in epidermal leaf cells of *N. benthamiana* and BRZ-treatment. The coloured data points represent the technical replicates of 2 independent experiments. The indicated p-values were obtained using a two-tailed heteroscedastic *t*-test and a Bonferroni multiple hypothesis correction. The presented images were acquired using confocal laser scanning microscopy (CLSM). Scale bars in (**C**) and (**G**) represent 5 μm, scale bars in (**D**) and (**H**) represent 2 μm. The areas that correspond to the images (**D**) and (**H**) are indicated by the dashed squares in images (**C**) and (**G**). Red arrowheads indicate endosomal compartments of BRI1-GFP. Endosomal compartments were omitted for quantitative analysis. The colour bar represents the colour code for fluorescence intensities.

The following figure supplement is available for figure 5:

**Figure supplement 1.** Control experiments for verifying the specific localisation patterns of FLS2 and BRI1.

other SERKs) compromises its function in immune signalling (*Ntoukakis et al., 2011*) we decided to omit these co-receptors for our study. Instead, we visualised FLS2 and BRI1 in complex with BSK1 or BIK1 using CLSM. As shown in *Figure 8* and *Figure 9*, in addition to the ligand binding receptors, the two RLCKs also appeared heterogeneously distributed, and BSK1 and BIK1 clusters became evident.

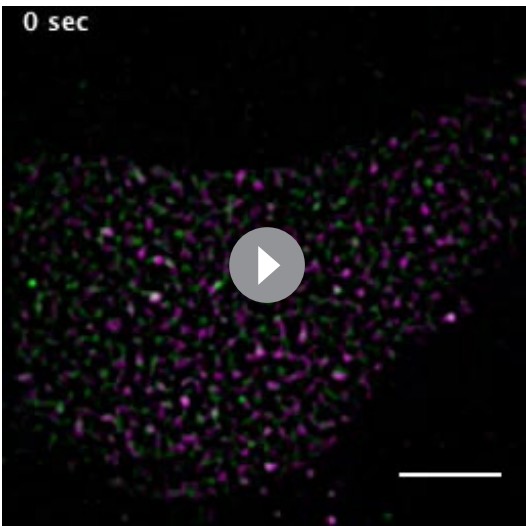

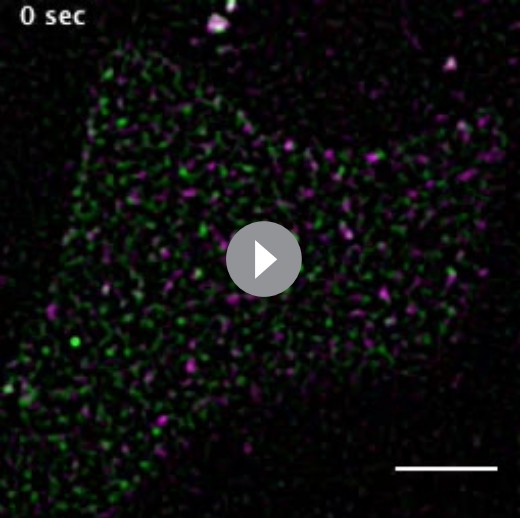

**Video 3.** Visualization of FLS2 receptor cluster dynamics within the plasma membrane. The presented time series was acquired from an epidermal leaf cell after transient co-expression of FLS2-GFP and FLS2-mCherry in *N. benthamiana* using variable angle epi-fluorescence microscopy (VAEM). The acquisition time was 0.25 s per frame per channel over 200 frames in total. FLS2-GFP fluorescence is shown in green, FLS2-mCherry fluorescence is shown in magenta. The scale bar represents 5 µm.

**Video 4.** Simultaneous visualization of FLS2 and BRI1 receptor cluster dynamics within the plasma membrane. The presented time series was acquired from an epidermal leaf cell after transient co-expression of BRI1-GFP and FLS2-mCherry in *N. benthamiana* using variable angle epi-fluorescence microscopy (VAEM). The acquisition time was 0.25 s per frame per channel over 200 frames in total. BRI1-GFP fluorescence is shown in green, FLS2-mCherry fluorescence is shown in magenta. The scale bar represents 5 µm.

In agreement with the localisation patterns of BSK1 and BIK1 alone, also BiFC complexes formed clusters within the PM (*Figure 8B and F*, *Figure 9B and F*). Quantitative image analysis and visual image inspection indicated independent cluster populations but similar co-localisation behaviour for BSK1 in complex with FLS2 and BRI1 (*Figure 8I*). Thus, we did not reveal major differences for FLS2 and BRI1 signalling complexes at this stage.

This situation changed dramatically when we focused our attention on BIK1 complexes. Although quantitative image analysis did not indicate differences between FLS2- and BRI1-BIK1 complexes (*Figure 9O*), localisation patterns were entirely different for immune and growth signalling complexes. As shown in *Figure 9*, BRI1-BIK1, but not FLS2-BIK1, complexes co-localised with microtubule-associated BIK1 populations (*Figure 9—figure supplement 1*). The finding that signalling complexes also localised differentially further underlined our notion of spatial separation for the two LRR-RKs within the PM of plant cells, in addition to the aforementioned differences in PM localisation for FLS2 and BRI1 themselves.

In conclusion, using various different imaging approaches we were able to discriminate specific PM localisation patterns for the PRR FLS2 and the hormone receptor BRI1. We showed that both ligand-binding receptors are heterogeneously distributed due to the formation of transient receptor clusters. We observed that FLS2 and BRI1 clusters undergo different dynamics but that ligand availability reduces lateral cluster mobility in both cases. Moreover, we could visualise receptor clusters for both LRR-RKs within distinct PM nanodomains and differential PM localisation for steady-state FLS2- and BRI1 signalling complexes. Taken together, our findings emphasise a model of signalling pathway-specific pools of downstream components and suggest that spatial separation of immune and growth signalling platforms may contribute to the generation of signalling specificity.

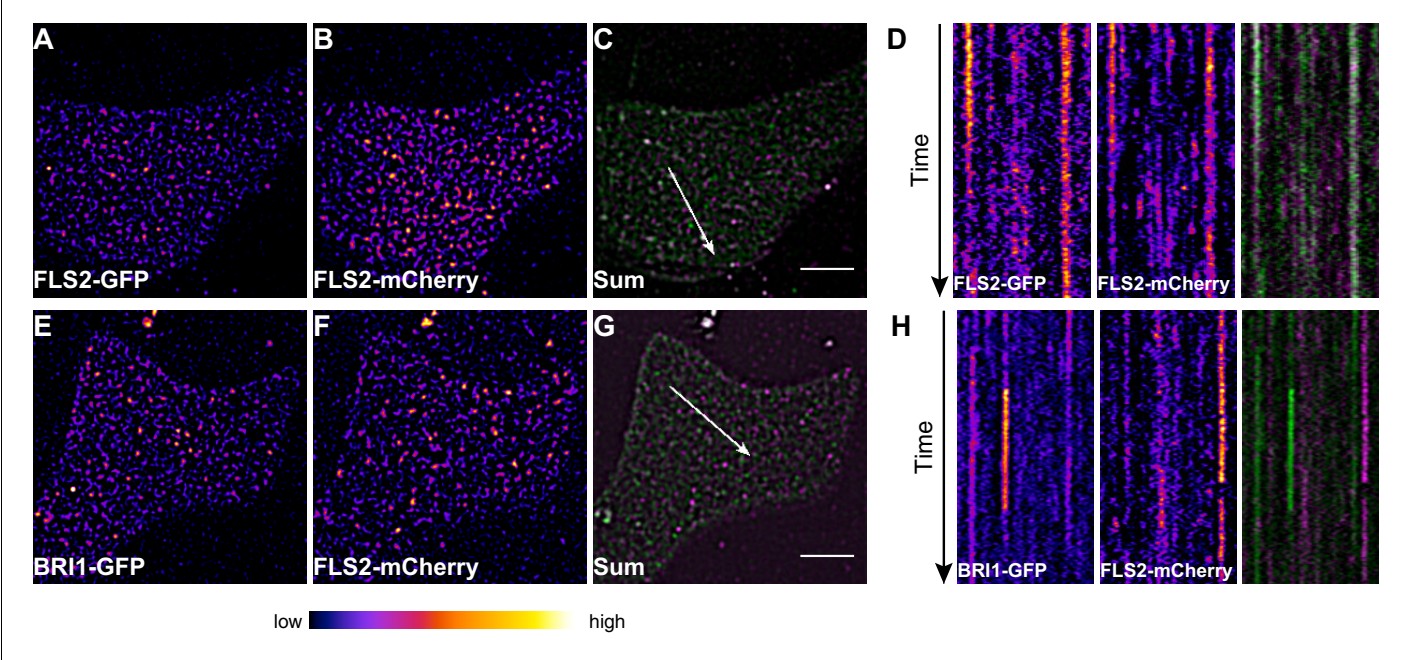

**Figure 6.** FLS2 and BRI1 clusters are spatiotemporally separated. (A–C) Plasma membrane localisation of FLS2-GFP (A) and FLS2-mCherry (B) after transient co-expression in epidermal leaf cells of *N. benthamiana* as well as the merged image (C). (D) Kymograph analysis of VAEM micrograph shown in (C). The spatial dimension of the kymograph is indicated by the white arrow in (C). The acquisition time of a single channel was 0.25 s. For each channel 200 frames were collected. (E–G) Plasma membrane localisation of BRI1-GFP (E) and FLS2-mCherry (F) after transient co-expression in epidermal leaf cells of *N. benthamiana* as well as the merged image (G). (H) Kymograph analysis of VAEM micrograph shown in (G). The spatial dimension of the kymograph is indicated by the white arrow in (C). The acquisition time of a single channel was 0.25 s. For each channel 200 frames were collected. The presented images were acquired using variable angle epi-fluorescence microscopy (VAEM). In the merged images FLS2-GFP or BRI1-GFP signals are shown in green and FLS2-mCherry signals are shown in magenta. The scale bar represents 5 µm. The colour bar represents the colour code for fluorescence intensities. Two independent experiments with similar results were performed.

## Discussion

Plants and animals employ cell surface receptors for the perception of various extracellular signals like hormones or PAMPs. In Arabidopsis, the immune receptor FLS2 and the growth receptor BRI1 represent two of the best-studied transmembrane receptors. Genetic and biochemical studies have linked extracellular ligand binding to the generation of specific flg22- or BR-triggered physiological responses. However, little is known about the organisation of these two receptors within the PM, the compartment where both LRR-RKs perceive their ligands and initiate their signalling cascades. Here, we provide evidence that FLS2 and BRI1 are heterogeneously distributed within the PM by formation of distinct PM nanodomain-localised receptor clusters. In addition, we show that the RLCKs BSK1 and BIK1 also form PM clusters, and that FLS2- and BRI1-BIK1 complexes localise differentially.

Recently, *Somssich et al. (2015)* reported a comparative study that investigated PM-associated signalling principles of plant RKs. Using the flg22- and CLAVATA3 (CLV3)-triggered pathways as examples for plant immune and growth signalling, they addressed the receptor/co-receptor association prior and during signalling initiation. Similar to previous studies (*Chinchilla et al., 2007*; *Heese et al., 2007*; *Schulze et al., 2010*; *Roux et al., 2011*; *Sun et al., 2013*), *Somssich et al. (2015)* revealed that the initiation of FLS2-mediated signalling strictly relies on ligand-induced complex formation between FLS2 and SERKs. In contrast, CLV-mediated signalling was found to depend on preformed receptor/co-receptor complexes (*Somssich et al., 2015*), in line with findings for the BRI1-BAK1 association in Arabidopsis roots (*Bücherl et al., 2013*). Interestingly, the association between receptors and co-receptors, but also BAK1 and its negative regulator BIR2 (*Halter et al., 2014*), occurs in clusters or subdomains of the PM, further emphasising our observations presented here.

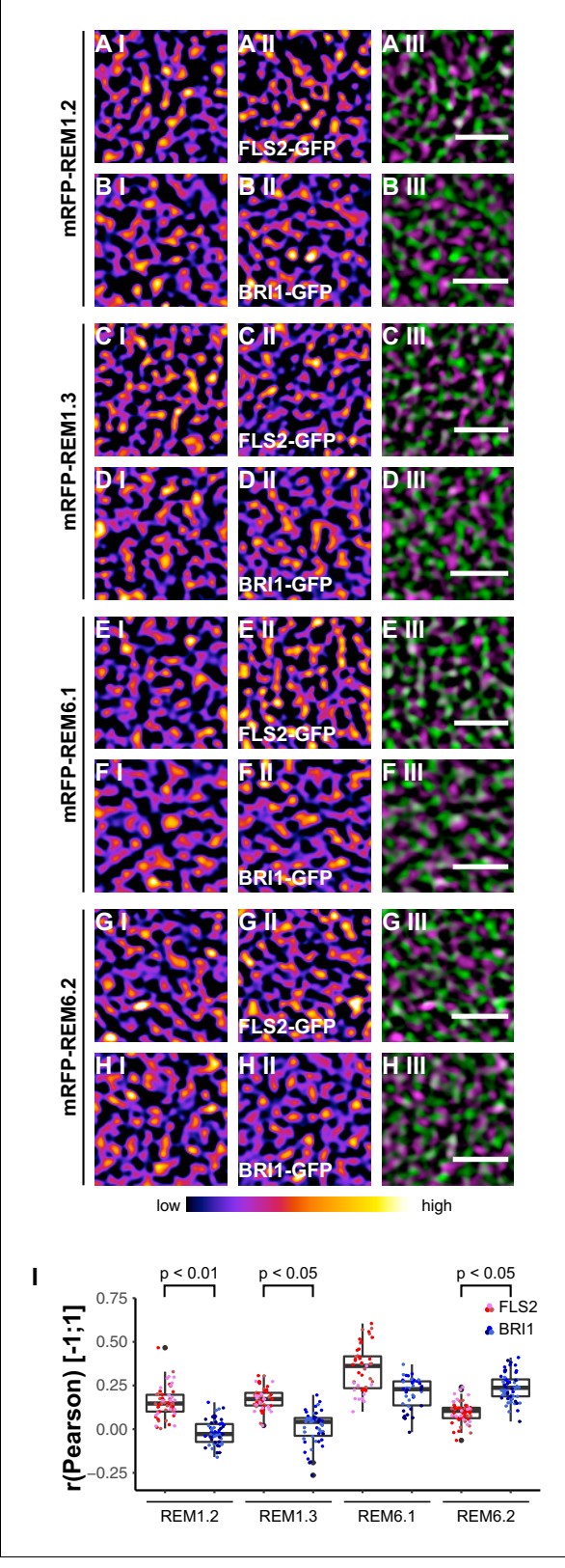

**Figure 7.** FLS2 and BRI1 co-localize differentially with remorin markers. (**A I–A III**) Confocal micrographs of mRFP-REM1.2 (**A I**) and FLS2-GFP (**A II**) plasma membrane localisation after transient co-expression in epidermal leaf cells of *N. benthamiana* as well as the merged image (**A III**). (**B I–B III**) Confocal micrographs of mRFP-REM1.2 (**B I**) and BRI1-GFP (**B II**) plasma membrane localisation after transient co-expression in epidermal leaf cells of *N.*

*Figure 7 continued on next page*

*Figure 7 continued*

*benthamiana* as well as the merged image (**B III**). (**C I–C III**) Confocal micrographs of mRFP-REM1.3 (**C I**) and FLS2-GFP (**C II**) plasma membrane localisation after transient co-expression in epidermal leaf cells of *N. benthamiana* as well as the merged image (**C III**). (**D I–D III**) Confocal micrographs of mRFP-REM1.3 (**D I**) and BRI1-GFP (**D II**) plasma membrane localisation after transient co-expression in epidermal leaf cells of *N. benthamiana* as well as the merged image (**D III**). (**E I–E III**) Confocal micrographs of mRFP-REM6.1 (**E I**) and FLS2-GFP (**E II**) plasma membrane localisation after transient co-expression in epidermal leaf cells of *N. benthamiana* as well as the merged image (**E III**). (**F I–F III**) Confocal micrographs of mRFP-REM6.1 (**F I**) and BRI1-GFP (**F II**) plasma membrane localisation after transient co-expression in epidermal leaf cells of *N. benthamiana* as well as the merged image (**F III**). (**G I–G III**) Confocal micrographs of mRFP-REM6.2 (**G I**) and FLS2-GFP (**G II**) plasma membrane localisation after transient co-expression in epidermal leaf cells of *N. benthamiana* as well as the merged image (**G III**). (**H I–H III**) Confocal micrographs of mRFP-REM6.2 (**H I**) and BRI1-GFP (**H II**) plasma membrane localisation after transient co-expression in epidermal leaf cells of *N. benthamiana* as well as the merged image (**H III**). (**I**) Quantitative co-localisation analysis for FLS2-GFP and BRI1-GFP with mRFP-REM1.2, mRFP-REM1.3, mRFP-REM6.1, and mRFP-REM6.2, respectively. The coloured data points represent the technical replicates of 3 independent experiments. The indicated p-values were obtained using a two-tailed heteroscedastic *t*-test and a Bonferroni multiple hypothesis correction. In the merged images FLS2-GFP or BRI1-GFP signals are shown in green and REM signals are shown in magenta. Scale bars represent 2 µm. The colour bar represents the colour code for fluorescence intensities. Black dots represent outliers.

The following figure supplement is available for figure 7:

**Figure supplement 1.** Control experiments for verifying the specific co-localisation of FLS2 and BRI1 with remorin markers.

---

Formation of PM receptor clusters is a well-known phenomenon for mammalian receptors like T-cell receptors (TCRs) or EGFR (*Dinic et al., 2015*; *Gao et al., 2015*; *Su et al., 2016*). The RTK EGFR constitutively undergoes clustering and EGFR clusters play important roles in cell signalling (*Clayton et al., 2007*; *Gao et al., 2015*; *Paviolo et al., 2015*). Using super-resolution microscopy, *Gao et al. (2015)* showed that EGFR clusters localise to PM nanodomains and that cluster formation relies on the integrity of the PM lipidome since sterol depletion of the PM using methyl-$\beta$-cyclodextrin (M$\beta$CD) disrupts clustering. Similar observations have recently been made for BRI1 in Arabidopsis roots (*Wang et al., 2015*). *Wang et al. (2015)* observed that BRI1 localises to FLOT1- and clathrin heavy chain (CHC)-labelled PM nanodomains and that partitioning of BRI1 into PM nanodomains as well as the PM lipidome are crucial for BR responses. These findings are in line with our observations for FLS2 and BRI1 in epidermal leaf cells since M$\beta$CD treatment also affected FLS2 clusters (*Figure 2—figure supplement 1*). Thus, both LRR-RKs constitutively formed PM receptor clusters that were influenced by the PM lipid-composition and localised to PM nanodomains labelled by distinct REM protein markers. However, the differential co-localisation to specific PM nanodomains allowed us to discriminate FLS2 and BRI1 receptor clusters, thus indicating their spatial separation within the PM.

Clustering of PM receptors is thought to provide a mechanism for modulating intermolecular interactions and for fine-tuning signal transduction (*Bray et al., 1998*; *Abulrob et al., 2010*). For example, *Hsieh et al., 2010* reported that EGFR clustering enhances the recruitment of downstream signalling components. Interestingly, we not only observed cluster formation for FLS2 and BRI1, but also for their downstream signalling components BSK1 and BIK1, as well as for the respective RK-RLCK complexes. We propose that similar mechanisms as described for EGFR may also account for FLS2- and BRI1-mediated signalling, and, furthermore, that clustering of PM proteins is a more general organising principle for plant PM proteins (*Kleine-Vehn et al., 2011*; *Wang et al., 2013*; *Demir et al., 2013*; *Jarsch et al., 2014*).

In fact, in animal cells, most lipids and proteins are heterogeneously distributed across the PM due to intermolecular interactions that generate inhomogeneities of varying size and stability (*Jaqaman and Grinstein, 2012*; *Saka et al., 2014*). Besides molecular interactions among PM constituents, the cortical cytoskeleton influences the formation and/or stability of PM nanodomains in animals and plants (*Plowman et al., 2005*; *Chichili and Rodgers, 2009*; *Jaumouillé et al., 2014*; *Dinic et al., 2013*; *Szymanski et al., 2015*; *Su et al., 2016*). Intriguingly, we repeatedly observed

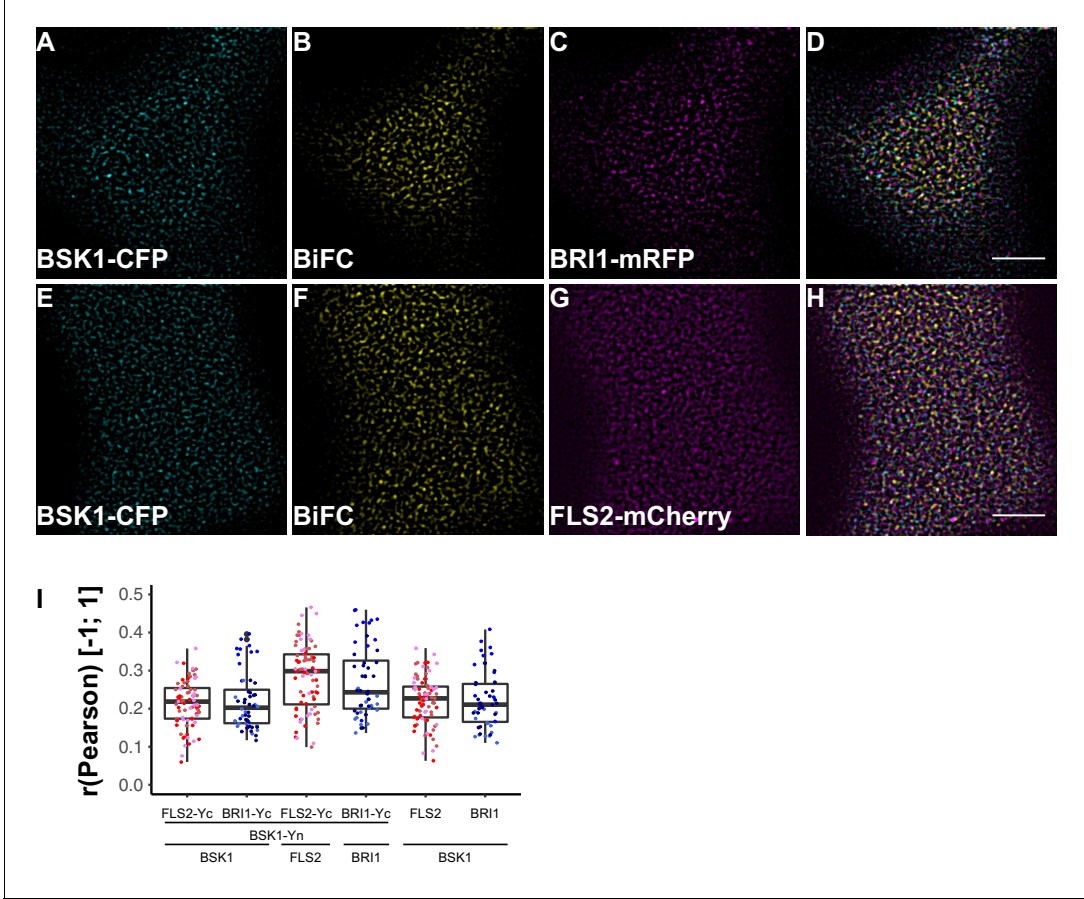

**Figure 8.** FLS2 and BRI1 signaling complexes also undergo cluster formation within the plasma membrane. (A–D) Confocal micrographs of BSK1-CFP (A), BSK1-nYFP/BRI1-cYFP (BiFC) (B), and BRI1-mRFP (C) plasma membrane localisation after transient co-expression in epidermal leaf cells of *N. benthamiana* as well as the merged image (D). (E–H) Confocal micrographs of BSK1-CFP (E), BSK1-nYFP/FLS2-cYFP (BiFC) (F), and FLS2-mCherry (G) plasma membrane localisation after transient co-expression in epidermal leaf cells of *N. benthamiana* as well as the merged image (H). (I) Quantitative co-localisation analysis for the reconstituted YFP fluorescence intensities (BiFC) with BSK1-CFP (BSK1) as well as BRI1-mRFP (BRI1) or FLS2-mCherry (FLS2) and the quantified co-localisation between BSK1-CFP with BRI1-mRFP or FLS2-mCherry. The coloured data points represent the technical replicates of 3 independent experiments. No statistical differences were observed based on two-tailed heteroscedastic *t*-tests and a Bonferroni multiple hypothesis correction. BiFC stands for bimolecular fluorescence complementation and the labelled image panels show YFP fluorescence signals for the respective protein complexes. Yc and Yn indicate the C- and N-terminal fragments of the split YFP fluorophore, respectively. Scale bars represent 5 µm. Black dots represent outliers.

close association between FLS2 and BRI1 clusters with actin filaments. Though, it is currently unclear how or whether actin contributes to the formation and/or stability or FLS2 and BRI1 clusters. It was however shown previously that actin-myosin function is required for flg22-induced endocytosis (*Beck et al., 2012*).

In addition to a contribution to PM nanodomain formation, cortical cytoskeleton components also affect lateral mobility of PM proteins in animal cells (*Chichili and Rodgers, 2009*; *Jaqaman and Grinstein, 2012*). However, in plant cells, it seems that the cell wall is mainly responsible for restricting movements of PM proteins within the lipid bilayer (*Martinière et al., 2012*). Similar to the findings of *Martinière et al. (2012)*, we observed very limited dynamics of FLS2 and BRI1 clusters within the PM. The study of *Jarsch et al. (2014)*, which described various different localisation patterns for the REM protein family in plant PMs, revealed that the clusters of these PM-associated proteins also hardly undergo lateral movements. Since REMs bind to the inner leaflet of PMs (*Konrad et al., 2014*) and therefore cannot directly interact with the extracellular cell wall, a restrictive influence of the cortical cytoskeleton on the lateral PM mobility should not be excluded.

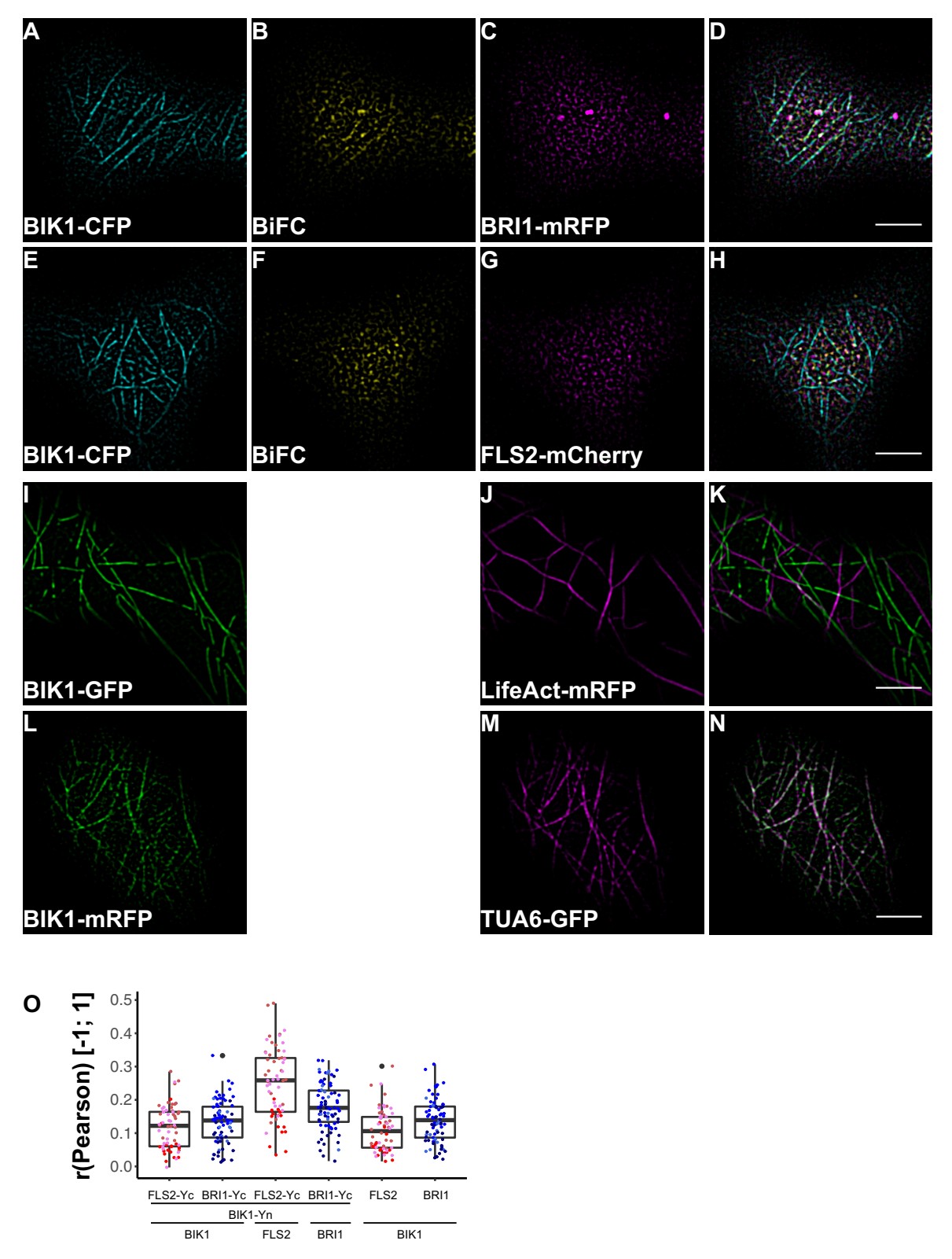

**Figure 9.** BRI1-BIK1, but not FLS2-BIK1, complexes associate with cortical microtubules. (A–D) Confocal micrographs of BIK1-CFP (A), BIK1-nYFP/BRI1-cYFP (BiFC) (B), and BRI1-mRFP (C) plasma membrane localisation after transient co-expression in epidermal leaf cells of *N. benthamiana* as well as the merged image (D). (E–H) Confocal micrographs of BIK1-CFP (E), BIK1-nYFP/FLS2-cYFP (BiFC) (F), and FLS2-mCherry (G) plasma membrane localisation after transient co-expression in epidermal leaf cells of *N. benthamiana* as well as the merged image (H). (I–K) Confocal micrographs of BIK1-GFP (I) and

*Figure 9 continued on next page*

*Figure 9 continued*

LifeAct-tRFP (J) fluorescence intensities after transient co-expression in epidermal leaf cells of *N. benthamiana* as well as the merged image (K). (L–N) Confocal micrographs of BIK1-mRFP (L) and TUB5-GFP (M) fluorescence intensities after transient co-expression in epidermal leaf cells of *N. benthamiana* as well as the merged image (N). (O) Quantitative co-localisation analysis for the reconstituted YFP fluorescence intensities (BiFC) with BIK1-CFP (BIK1) as well as BRI1-mRFP (BRI1) or FLS2-mCherry (FLS2) and the quantified co-localisation between BIK1-CFP with BRI1-mRFP or FLS2-mCherry. The coloured data points represent the mean values of 3 independent experiments. No statistical differences were observed based on two-tailed heteroscedastic *t*-tests and a Bonferroni multiple hypothesis correction. BiFC stands for bimolecular fluorescence complementation and the labelled image panels show YFP fluorescence signals for the respective protein complexes. Yc and Yn indicate the C- and N-terminal fragments of the split YFP fluorophore, respectively. LifeAct-tRFP was employed to visualise actin filaments, whereby tRFP stands for TagRFP. Scale bars represent 5 µm. Black dots represent outliers.

The following figure supplement is available for figure 9:

**Figure supplement 1.** BIK1 and BRI1-BIK1 complexes associate with cortical microtubules.

Our quantification of FLS2 and BRI1 cluster densities and sizes yielded similar values as described by *Jarsch et al. (2014)* for REM nanodomains. With around two clusters per µm$^2$, the detected cluster densities were slightly higher than determined for the REM nanodomain markers, which range from 0.1 to 1 per µm$^2$ (*Jarsch et al., 2014*). Although we revealed cluster diameters in the range of approximately 250–500 nm that are again in line with the dimensions of REM PM nanodomains (*Jarsch et al., 2014*), we assume that the actual size of FLS2 and BRI1 clusters is smaller. Similar to our observations, *Demir et al. (2013)* reported PM nanodomain sizes of ca. 250 nm for the potato REM1.3 when visualized by CLSM. However, subsequent analysis of these protein clusters using the super-resolution imaging method STED (stimulated emission depletion) led to more confined dimensions of around 100 nm (*Demir et al., 2013*).

Even though the spatial features of FLS2 and BRI1 clusters were comparable and similar to REM proteins, we observed major differences for the dynamics of protein clusters, both among the transmembrane receptors and with regard to REM proteins. In contrast to the high stability of REM nanodomains (*Jarsch et al., 2014*), the lifetimes of FLS2 and BRI1 clusters were much shorter and they exhibited a more dynamic behaviour. Furthermore, comparison between FLS2 and BRI1 populations revealed that the PRR clusters were characterized by increased stability under steady-state conditions. The low lateral mobility of the two LRR-RK cluster populations and the observation of subpopulations with higher displacement values are in line with the recent report of *Wang et al. (2015)*. They described two subpopulations for BRI1 with low and high lateral mobility, respectively, in the PM of epidermal root cells. Under steady-state conditions, the majority of BRI1 receptors underwent only short-range movements (*Wang et al., 2015*), similar to our results for epidermal leaf cells. In stark contrast to *Wang et al. (2015)* are our observations of reduced cluster mobility for FLS2 and BRI1 receptors in the presence of their respective ligands. However, our observations are in line with a previous report about FLS2 in Arabidopsis protoplasts (*Ali et al., 2007*) and with findings for the mammalian PM receptor EGFR (*Low-Nam et al., 2011*). Thus, increased receptor confinement in response to ligand binding may be a more general phenomenon for cell surface receptors. A plausible explanation for reduced lateral mobility is the well-established formation or stabilisation of receptor (hetero-)oligomers within the PM for activation of signal transduction as previously reported for FLS2 and BRI1 (*Chinchilla et al., 2007*; *Somssich et al., 2015*; *Wang et al., 2008*; *Bücherl et al., 2013*). The low lateral mobility of PM proteins in general and the additional confinement in response to ligands also suggests a ligand-independent pre-organisation of receptors and/or signalling units (*Martinière et al., 2012*; *Abulrob et al., 2010*; *Sandor et al., 2016*).

*Wang et al. (2015)* also showed that BRI1 receptors co-localise with the PM nanodomain marker FLOT1. Since proteomic studies previously revealed that REM1.2 and REM1.3 are phosphorylated and enriched in detergent-resistance membranes in a flg22-dependent manner (*Benschop et al., 2007*; *Keinath et al., 2010*), we used several REM proteins as references for investigating the localisation of FLS2 and BRI1 to PM nanodomains. The identified differential co-localisation of the two transmembrane receptors with four different REM markers clearly demonstrated that both FLS2 and BRI1 clusters reside within PM nanodomains. Combined with our hypothesis of an interplay between

receptor clusters and the cortical cytoskeleton, these findings are in agreement with the three-tiered hierarchical PM organization model proposed by *Kusumi et al. (2011)*.

Moreover, the visualisation of the two LRR-RKs in different PM nanodomains underlined our results obtained by FLS2-BRI1 co-localisation that indicated a spatial separation for immune and growth receptors in plant PMs in steady-state conditions. The visualisation of FLS2 and BRI1 signalling complexes additionally confirmed these results. In particular, we found that BRI1-BIK1, but not FLS2-BIK1, complexes associated with cortical microtubules. Cortical microtubules are also anchorpoints for the cellulose synthase complex (*Paredez et al., 2006*; *Gutierrez et al., 2009*). Intriguingly, activation of BR signalling has an immediate effect on cell wall morphology (*Elgass et al., 2009*). Thus, a link between microtubule-associated BRI1-BIK1 complexes and BR-induced cell wall changes may exist. Interestingly, BIK1 fulfils opposing roles for BR and immune signal transduction (*Lin et al., 2013*). Whereas BIK1 is a positive regulator of FLS2-mediated signalling by connecting, for example, ligand perception to ROS production (*Kadota et al., 2014*; *Li et al., 2014*), BIK1 negatively regulates BRI1-mediated cellular responses (*Lin et al., 2013*). Based on our observations, this differential signalling specificity may be encoded in differential protein complex localisation (and composition) at the PM.

Only 5–10% of BRI1 receptor molecules are actively involved in BR signal transduction (*van Esse et al., 2012*). In addition, hyper-activation of BRI1 by exogenous application of BRs results in only a partial mobility shift of BRI1, whereas its chemical or genetic inactivation only affects a subpopulation of BRI1 receptors (*Wang et al., 2015*). Interestingly, the observation that only a subset of FLS2 receptors are present in steady-state complexes with their downstream substrates BIK1 and BSK1, in what could be defined as pre-formed signalling platforms, correlates with the previous observation that flg22 binding to only a small subset of receptors is required for inducing a saturating response (*Meindl et al., 2000*; *Bauer et al., 2001*).

Collectively, our results suggest a spatial separation of FLS2 and BRI1 signalling platforms under steady-state conditions. These findings demonstrate the existence of pathway-specific signalling component pools as previously postulated (*Albrecht et al., 2012*; *Halter et al., 2014*). The establishment of physically separated signalling units is an economically favourable concept for plant cells, since common signalling components can be employed along different and even antagonistic signal transduction routes. We propose that the spatiotemporal separation of FLS2 and BRI1 signalling platforms in steady-state conditions is a means to generate signalling specificity between the two pathways upon ligand perception, in addition to potential differential phosphorylation of common signalling components (*e.g.* SERK3/BAK1, BIK1, and BSK1). Whether differential phosphorylation of common signalling components in fact is a consequence of spatially separated signalling complexes will be investigated in future studies, besides the composition and stoichiometry of distinct PM signalling pools.

## Methods and materials

### Plant materials

Arabidopsis seedlings were grown on Murashige and Skoog (MS) salt medium containing 1% sucrose and 0.8% agar with a 16 hr photoperiod at 22°C. *N. benthamiana* plants were soil-grown under a photoperiod of 16 hr and at 22°C.

We used the previously published Arabidopsis lines Col-0/pFLS2::FLS2-GFP (*Göhre et al., 2008*), Col-0/pBRI1::BRI1-GFP (*Geldner et al., 2007*), and Col-0/p35S::mCherry-TUA5 (*Gutierrez et al., 2009*). The double transgenic lines pBRI1::BRI1-GFP/p35S::mCherry-TUA5 and pFLS2::FLS2-GFP/p35S::mCherry-TUA5 were generated by crossing and imaged in the F1 population.

For the generation of the Col-0/p35S::Lifeact-TagRFP line, a previously described plasmid was used (*Tilsner et al., 2012*). Col-0 plants were stably transformed using the floral dip method (*Clough and Bent, 1998*). The double transgenic pBRI1::BRI1-GFP/p35S::Lifeact-TagRFP and pFLS2::FLS2-GFP/p35S::Lifeact-TagRFP lines were generated by crossing and imaged in the F1 population.

## Molecular cloning

The REM constructs used in this study were described previously (*Jarsch et al., 2014*). For the generation of the bimolecular fluorescence complementation (BiFC) constructs, BIK1-CFP, BIK1-mRFP, BRI1-GFP, BRI1-mRFP, and BSK1-CFP BIK1, BRI1, and FLS2 were PCR amplified from Arabidopsis Col-0 cDNA and BSK1 was PCR amplified from Arabidopsis Col-0 genomic DNA. Subsequently, the gel purified PCR products were inserted via Gateway TOPO reaction into pENTR-D-TOPO plasmids and the products were verified via sequencing. Expression vectors for BiFC experiments were generated by Gateway LR reactions using the respective pENTR clones and the pAM-35S-GW-YFPc as well as pAM-35S-GW-YFPn (*Lefebvre et al., 2010*), CFP-tagged constructs using pGWB44 (*Nakagawa et al., 2007*), RFP-tagged using pB7RWG2 (*Karimi et al., 2002*), GFP-tagged constructs using pK7FGW2 (*Karimi et al., 2002*). FLS2-mCherry was generated from pCAMBIA2300 pFLS2:: FLS2-3xMYC-GFP. Using the SalI restriction enzyme and subsequent ligation, GFP was substituted with mCherry.

## Transient expression in *N. benthamiana*

*Agrobacterium tumefaciens* GV3101 or GV3103 strains carrying p35S::BIK1-CFP, p35S::BSK1-CFP, pFLS2::FLS2-GFP, p35S::BRI1-GFP, p35S::BIK1-YFPn, p35S::BSK1-YFPn, p35S::BRI1-YFPc, p35S:: FLS2-YFPc, p35S::BRI1-mRFP, pFLS2::FLS2-mCherry, p35S::GFP-TUA6, or p35S::Lifeact-TagRFP were grown overnight in Luria-Bertani (LB) medium containing the respective antibiotics. Bacteria were harvested by centrifugation at 3.000 g for 5 min and re-suspended in $H_2O$. After a second centrifugation step bacteria were re-suspended in infiltration buffer containing 10 mM $MgCl_2$, 10 mM MES pH 5.6, and 100 µM acetosyringone. Bacterial suspensions were adjusted to $OD_{600} = 0.2$ or, for bimolecular fluorescence complementation constructs containing suspensions, to $OD_{600} = 0.4$. Subsequently, the respective bacterial suspensions were combined for co-infiltration with a bacterial suspension providing the p19 helper plasmid. The final *Agrobacterium tumefaciens* suspension mixes were syringe infiltrated into the adaxial side of *N. benthamiana* leaves. The used *N. benthamiana* plants were 4–5 weeks old.

## Confocal laser scanning microscopy (CLSM)

Cotyledons of Arabidopsis seedlings or leaf discs of *N. benthamiana* were imaged on a Leica TCS SP5 (Leica, Germany) confocal microscope using a 63 × 1.2 NA water immersion objective. CFP, GFP, and YFP were excited using the Argon ion laser lines 458 nm, 488 nm, and 514 nm, respectively. TagRFP, mRFP and mCherry were excited using a DPSS laser (561 nm) when imaged in combination with GFP or using a HeNe laser (594 nm) when imaged in combination with YFP. Fluorescence emission was collected within following band width generated by an AOTF: 465–505 nm for CFP, 500–540 nm for GFP, 520–560 nm for YFP, and 590–630 nm (DPSS excitation) or 600–640 nm (HeNe excitation) for TagRFP/mRFP/mCherry.

## Variable angle epifluorescence microscopy (VAEM)

Cotyledons of Arabidopsis seedlings or leaf discs of *N. benthamiana* were imaged on a Zeiss Elyra PS1 (Zeiss, Germany) microscope using a 100 × 1.4 NA oil immersion objective. GFP was excited using a 488 nm solid-state laser diode and mCherry was excited using a 561 nm solid-state laser diode. Fluorescence emission was collected with an EM-CCD camera with bandwidth filters ranging from 495–550 nm for GFP and 575–635 nm for mCherry. For samples expressing GFP only fluorescence was collected with an acquisition time of 500 ms. For GFP/mCherry expressing samples the acquisition times were 250 ms for each channel using the 'fast-channel' mode. Throughout image acquisition the microscope was operated in the 'TIRF' mode.

## Plant treatments

For brassinazole (BRZ; TCI, UK) treatment, 20 mM DMSO stock solution was diluted to a final concentration of 5 µM BRZ using liquid MS medium. Arabidopsis seedlings were transferred 3 days post germination in a 12-well plate containing the 5 µM BRZ solution or mock, respectively, and cultivated for another 2 days prior to imaging. For treatment of *Nicotiana benthamiana* leaves, BRZ was added to the infiltration suspension in a final concentration of 10 µM and co-infiltrated.

For 24-epi-brassinolide (BL; Sigma-Aldrich, UK) treatment, 1 mM EtOH stock solution was diluted to a final concentration of 100 nM BL using liquid MS medium. For microscopy cotyledons were directly mounted in 100 nM BL and incubated for the indicated time period.

For flg22 treatment, an aqueous 100 µM stock solution of flg22 (EZ Biolabs, USA) was diluted to a final concentration of 100 nM flg22 using liquid MS medium. For microscopy, cotyledons were directly mounted in 100 nM flg22 and incubated for the indicated time period.

For methyl-$\beta$-cyclodextrin (M$\beta$CD; Sigma-Aldrich, UK) treatment, a 30 mM M$\beta$CD solution was prepared in liquid MS. Plants were incubate for the indicated time period in a 12-well plate at room temperature prior to mounting in the same solution on object slides for microscopy.

## Image processing

Confocal and VAEM micrographs were analysed and modified using FIJI (ImageJ 2.0.0–39/rc-1.50b). To emphasise cluster formation, the presented images and time series were modified using the 'LoG3D' plugin (*Sage et al., 2005*). Time series were additionally neutralised with a saturation of zero prior to contrasting. For the generation of kymographs, we used the plugin 'Multi Kymograph' with a line width of one.

## Particle detection and analysis

For single particle analysis we used the plugin TrackMate (2.7.4). After selecting a region of interest (ROI) encompassing the cell outline the LoG detector was selected. Based on preliminary particle analysis using FIJI the estimated blob diameter was set to 0.3 µm and the threshold was set to zero. To exclude false-positive particles and endomembrane compartments we applied a quality (Auto setting) and mean intensity filter. Subsequently, the simple LAP tracker was selected with a maximal linking distance of 0.5 µm and without gap-closing. Using the analysis option within the TrackMate dialog window we obtained statistics for particle size, track duration, and particle displacement.

## Co-localisation analysis

Co-localisation analysis was carried out as described previously (*Jarsch et al., 2014*) using ImageJ/FIJI. Endomembrane compartments were excluded from the co-localisation analysis by ROI selection.

Briefly, acquired confocal images were 'Mean' filtered with a radius of 2 pixels. Background was subtracted using the 'Rolling ball' method with a radius of 20 pixels. ROIs were manual selected and for quantifying the co-localisation the plugin 'Intensity Correlation Analysis' (*Li et al., 2004*) was applied.

## Statistical analysis

Statistical analysis using Student's t-tests was carried out in Numbers (Apple, USA). Box plots were generated in R.

## Accession numbers

BIK1 (AT2G39660), BRI1 (AT4G39400), BSK1 (AT4G35230), FLS2 (AT5G46330), REM1.2 (AT3G61260), REM1.3 (AT2G45820), REM6.1 (AT2G02170), REM6.2 (AT1G30320), TUA5 (AT5G19780), TUB5 (AT1G20010).

## Acknowledgements

The authors thank Jordi Chan for providing the 35S::GFP-TUA6 construct, Niko Geldner for providing the Col-0/pBRI1::BRI1-GFP line, Jelmer J Lindeboom for providing the Col-0/p35S::mCherry-TUA5 line, Jens Tilsner for providing the p35S::Lifeact-TagRFP plasmid, and Martina Beck for providing the Col-0/p35S::Lifeact-TagRFP line. The authors are also thankful to Grant Calder and Eva Wegel (JIC Bioimaging facility) for assistance at the microscopes. This research was funded by the Gatsby Charitable Foundation (CZ; SR; DM), the European Research Council (grant 'PHOSPHinnATE') (CZ), as well as an Emmy-Noether grant, the Collaborative Research Center 924 (SFB924) both of the Deutsche Forschungsgemeinschaft (DFG) (TO), and Universität Bayern (IKJ).

# Additional information

## Funding

| Funder | Author |
|---|---|
| Gatsby Charitable Foundation | Silke Robatzek<br>Daniel MacLean<br>Cyril Zipfel |
| H2020 European Research Council | Cyril Zipfel |
| Deutsche Forschungsge-meinschaft | Thomas Ott |
| Universitat Bayern | Iris K Jarsch |

The funders had no role in study design, data collection and interpretation, or the decision to submit the work for publication.

## Author contributions

CAB, Conceptualization, Data curation, Formal analysis, Validation, Investigation, Visualization, Methodology, Writing—original draft, Writing—review and editing; IKJ, TO, Resources, Software, Visualization, Writing—review and editing; CSc, Data curation, Software, Formal analysis, Visualization; CSe, MM, SR, Resources, Writing—review and editing; DM, Data curation, Software, Visualization, Writing—review and editing; CZ, Conceptualization, Supervision, Funding acquisition, Investigation, Writing—original draft, Project administration, Writing—review and editing

## Author ORCIDs

Christian Schudoma, http://orcid.org/0000-0003-1157-1354
Cécile Segonzac, http://orcid.org/0000-0002-5537-7556
Daniel MacLean, http://orcid.org/0000-0003-1032-0887
Cyril Zipfel, http://orcid.org/0000-0003-4935-8583

# Additional files

## Supplementary files

• Supplementary file 1. Summary of quantitative image analysis. In this table a summary of the quantitative image analysis is given providing the number of independent biological experiments, the number of technical replicates as well as the results of t-tests and the corresponding final p-values. The final p-values were obtained by multiplying the *t*-test results with a Bonferroni factor of 2. For the comparison of FLS2 and BRI1 two-tailed heteroscedastic *t*-tests were applied. For the comparison of original and rotated images two-tailed homoscedastic t-tests were applied. For time series experiments one-tailed heteroscedastic *t*-tests were applied. The results of t-tests and final p-values were based on the analysis of the corresponding mean values for respective independent biological experiments. The presented mean values are in the units shown in the respective figures. 'SD' stands for standard deviation based on the technical replicates. 'At' stands for *Arabidopsis thaliana* and 'Nb' stands for *Nicotiana benthamiana*. 'FLS2/BSK1', 'BRI1/BSK1', 'FLS2/BIK1', and 'BRI1/BIK1' represent the respective BiFC complexes.

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
