## [Decision Letter]

[Editors’ note: a previous version of this study was rejected after peer review, but the authors submitted for reconsideration. The first decision letter after peer review is shown below.]

Thank you for submitting your work entitled "Plant immune and growth receptors share common signaling components but localise to distinct plasma membrane nanodomains" for consideration by *eLife*. Your article has been reviewed by three peer reviewers, one of whom, Thorsten Nürnberger (Reviewer #1), is a member of our Board of Reviewing Editor, and the evaluation has been overseen by Detlef Weigel as the Senior Editor.

Our decision has been reached after a long consultation between the reviewers. Based on these discussions and the individual reviews below, we regret to inform you that your work will not be considered further for publication in *eLife*.

All referees agree that your research addresses a key problem in receptor biology, which is how common co-receptors serve different receptors that are implicated in different or even opposing physiological programs. They also agree that the type of cell biology approach chosen here is timely and potentially suited to address the experimental problem to be solved. However, the referees also raised a number of technical and conceptual concerns about your work that due to their substantial nature prohibit a more positive decision for the moment. In particular, criticism has been raised mainly because of the extensive use of the transient expression system, and that co-localization (or the absence of it) of BRI1-GFP/FLS2-RFP, BRI1-GFP/BAK1-RFP and FLS2-GFP/BAK1-RFP have not been firmly established in stable transgenics. Moreover, since endogenous brassinolide levels might influence cluster formation, it would have been desirable to see the experiment to be conducted either in BL-deficient lines or by use of brassinazole. This would have allowed to compare truly non-stimulated BRI1 and FLS2 pathways. Although the referees acknowledge that your work did not attempt to study the fate of individual receptor complexes post stimulation, there was consensus that such data would have substantially increased the impact of your study for the field and beyond. In the light of these concerns, I regret to say that we cannot be more supportive at this point. Should you feel able, however, to address the aforementioned issues in an appropriate way *eLife* would invite you to re-submit your revised work.

Reviewer #1:

In my view, this is an important piece of work reporting a phenomenon suited to explain how common co-receptors serve surface receptors that are implicated in physiologically diverse programs, such as immunity and development. As this appears to be the first study ever to address such topic in higher organisms I clearly see the novelty and why it would make a valuable contribution to *eLife*. As I cannot really comment the quality and validity of the cell biological work I support publication of this work because of its eminent importance for the field and beyond.

Reviewer #2:

FLS2 and BRI1 trigger different gene transcriptional profiles and physiological responses. However, they share common co-receptor SERKs. In addition, different members of RLCKs function differentially downstream of the RLK complexes. How the signaling specificity is governed at the RLK level is not clear. This work examined how FLS2 and BRI1 distribute on the PM mainly by imaging the GFP or mCherry-tagged proteins without ligand treatments. They concluded that FLS2 and BRI1 are in PM nanoclusters, majority of which are in a spatially not over-lapping manner at the steady state. FLS2, but not BRI1, co-localizes with specific remorin protein-markers for certain nanodomains. FLS2 and BRI1 are associated with the cytoskeleton with FLS2 linked to actin-myosin and BRI1 linked to cortical microtubules.

This work is interesting and provides some insights into the FLS2 and BRI1 signaling. It is known that upon the ligand perception, FLS2 and BRI1 complex with SERK3 and activate the signaling. However, the majority of work here was done with FLS2 without flg22 treatment and examines the steady state of FLS2. BRI1 localization was done in a similar fashion-without BR treatment. Unlike FLS2 signaling, the endogenous BR signaling is constantly on in the cells (although likely at a low level). It remains unknown how the steady-state FLS2 patterns are related to the ligand-activated clusters and how this change may affect the signaling specificity, the key question that this manuscript is attempting to address. Recent work reported a highly dynamic behavior of FLS2 and BRI1 in particular in response to the cognate ligand perception (Somssich et al., Real-time dynamics of peptide ligand-dependent receptor complex formation in planta. Science Signaling 2015; Wang et al., Spatiotemporal dynamics of the BRI1 receptor and its regulation by membrane microdomains in living Arabidopsis cells. Molecular Plant 2015). The conclusion that FLS2 and BRI1 are largely in spatially separated nanoclusters at the steady state was based on the comparison between FLS2 and BRI1 localization in the absence of ligands in WT plants. The BR signaling is on in the absence of exogenous BR treatment. What they have observed in this experimental setting may contain both steady-state and activated form of BRI1. Additional experiments with BR biosynthetic mutants will be helpful to confirm the conclusion. Comparison the RLK patterns with and without ligands will provide insights into how these patterns contribute the signaling specificity.

Considering that the nanodomains are usually 10-100 nm in size, the resolutions of current confocal microscopic techniques to differentiate the two objectives are rather limited (CLSM 500~700 nm). Cautious should be taken into account for the spatial separations of clusters. As only 5~10% BRI1 are actively involved in BR signaling (van Esse et al., 2012), the BRI1 patterns are based on the observations from the "majority" of BRI1. At the moment, the knowledge is lacking on where those 5~10% activated portions are. The highly dynamic nature of BRI1 and FLS2 clusters upon the ligand perception will add another layer of complications.

Reviewer #3:

Bücherl et al. report that the immune receptor FLS2 and the brassinosteroid receptor BRI1 are found in distinct nanodomains of the plasma membrane (PM) that display differences in mobility and association to the cytoskeleton and contain different remorins. The topic is exciting and timely as understanding the spatiotemporal dynamics of receptor interactions will be key to understanding how specificity is generated at the PM. Unfortunately this manuscript does not advance our understanding of this topic sufficiently to warrant publication in *eLife* and the quality of the data presented raises several points:

1) Based on the similarity in cluster size observed in stable transgenic Arabidopsis line and after transient expression in *N. benthamiana* all following experiments are performed in *N. benthamiana* epidermal cells. In remains unclear if the two signalling pathways are operational and if endogenous receptors could interfere with the analysis.

2) The Pearson correlation coefficients for FLS2-GFP and FLS2-mCherry are moderate at best. Jarsch et al. report values of around 0,7 for individual remorins tagged with GFP and mRFP and values below 0,5 as observed for FLS2-GFP and FLS2-mCherry would be sitting on the fence of random distribution and positive correlation according to Jarsch et al. This could either be a technical issue regarding alignment of the CLSM that can easily be controlled by using multicolour fluorescent beads. Alternatively the low correlation coefficient could indicate that one of the two fusion proteins is non-functional and thus behaves differently. Has functionality of the FLS2-fusions been fully evaluated by mutant complementation? Given that the authors excluded the co-receptor BAK1 form their analysis as it has been shown that functionality of BAK1-GFP is compromised, the authors are obviously aware of the importance of this matter but I could not find convincing evidence for full functionality of FL2-XFP in the published literature.

3) To go beyond the level of high-resolution descriptive the authors would need to include data supporting that the observed nanoclusters are functionally relevant. What happens to signalling in remorin mutants? Are nanoclusters dependent on the cytoskeleton? How is the co-receptor BAK1 distributed and does absence of BAK1 affect nanocluster formation?

[Editors’ note: what now follows is the decision letter after the authors submitted for further consideration.]

Thank you for resubmitting your work entitled "Plant immune and growth receptors share common signaling components but localise to distinct plasma membrane nanodomains" for further consideration at *eLife*. Your revised article has been favorably evaluated by Detlef Weigel (Senior editor) and three reviewers, one of whom, Thorsten Nürnberger, is a member of our Board of Reviewing Editors.

The manuscript has been improved but there are two remaining issues that need to be addressed before acceptance, as outlined below:

1) The Pearsson correlation coefficient for FLS2-GFP vs FLS2-mCherry is surprisingly low. You argue that this is due to the low number of receptors per cluster and the dynamic behaviour of the clusters. This is one possible explanation, yet in the absence of a positive control technical reasons cannot be excluded.

2) Regarding functionality of FLS2-GFP the authors refer to Mbengue et al., 2016 yet one does not find evidence in there that demonstrates a wt-level flg22 response in the complementation line used.

---

## [Author Response]

[Editors’ note: the author responses to the first round of peer review follow.]

All referees agree that your research addresses a key problem in receptor biology, which is how common co-receptors serve different receptors that are implicated in different or even opposing physiological programs. They also agree that the type of cell biology approach chosen here is timely and potentially suited to address the experimental problem to be solved. However, the referees also raised a number of technical and conceptual concerns about your work that due to their substantial nature prohibit a more positive decision for the moment. In particular, criticism has been raised mainly because of the extensive use of the transient expression system, and that co-localization (or the absence of it) of BRI1-GFP/FLS2-RFP, BRI1-GFP/BAK1-RFP and FLS2-GFP/BAK1-RFP have not been firmly established in stable transgenics.

We agree with the Reviewers that working with stable transgenic lines is always desirable. However, the generation of multiple tagged-lines that would be required for the multiple co- localization experiments as performed in our study would have been extremely tedious. Importantly, several recent excellent publications in the plant field have used heterologous transient expression of tagged *Arabidopsis* PM proteins to study their dynamic localization pattern using confocal microscopy (*e.g.* Martinière et al., 2012; Jarsch et al., 2014; Chaparro-Garcia et al., 2015; Bozkurt et al., 2015; Somssich et al., 2015; Mbengue et al., 2016). In addition, most, if not all, similar studies in animals (all of them being published in respected high-ranking journals) are also performed upon transient heterologous expression, for similar reasons.

It is also important to flag that we have provided a comparison of PM localization patterns of FLS2-GFP and BRI1-GFP expressed stably in Arabidopsis under the expression of their native promoter or transiently in *Nicotiana benthamiana* (Figure 1), which revealed that the localization of the respective receptors was similar in both systems. This justified our use of transient expression in *N. benthamiana* for further experiments presented in this manuscript. Also, the main aim of our study is to perform a side-by-side comparison of the localization patterns and properties of FLS2 and BRI1 under steady-state conditions; our results provide ample evidence that these are distinct (based also on co-localization with distinct REM markers and downstream RLCKs).

We nevertheless provide here some preliminary evidence for the co-localisation of BAK1- mCherry with FLS2-GFP and BRI1-GFP, respectively, within PM nanodomains of stable transgenic Arabidopsis lines (see Figure 10). Moreover, data included in a manuscript submitted elsewhere show BRI1/BAK1 association within such nanodomains (Hutten et al., under revision with PLoS One; co-last authors: Johannes Hohlbein & Jan Willem Borst; co-author: CA Bücherl). Unfortunately, we have been so far not able to address this question for the FLS2/BAK1 combination, due to technical feasibility regarding FLIM in living leaf tissue and our inability to detect red fluorescent when expressing FLS2-RFP in stable transgenic plants.

Author response image 1.Subpopulations of FLS2 and BRI1 co-localise with BAK1 in plasma membrane nanodomains.The images show the plasma membrane localisation of FLS2-GFP or BRI1-GFP with BAK1- mCherry. The white arrowheads indicate plasma membrane nanodomains that contain fluorescently labelled receptor and co-receptor molecules simultaneously. The images were acquired 5 days post germination using VAEM and the scale bars represent a distance of 5 µm. The lines were generated by crossing pFLS2::FLS2-GFP (Göhre et al., 2008) or pBRI1::BRI1- GFP (Friedrichsen et al., 2000), respectively, with pBAK1::BAK1-mCherry (Bücherl et al., 2013).**DOI:**
http://dx.doi.org/10.7554/eLife.25114.023

Moreover, since endogenous brassinolide levels might influence cluster formation, it would have been desirable to see the experiment to be conducted either in BL-deficient lines or by use of brassinazole. This would have allowed to compare truly non-stimulated BRI1 and FLS2 pathways.

We agree with the Reviewers that for full comparison the localisation of both receptors should also be assessed under similar activation states. Therefore, we now included experiments after BRZ treatment in the revised manuscript. The respective results are shown in current Figure 3 and Figure 5 (see also 1. Comment, section: reviewer #2).

Although the referees acknowledge that your work did not attempt to study the fate of individual receptor complexes post stimulation, there was consensus that such data would have substantially increased the impact of your study for the field and beyond.

We have now addressed the immediate impact of flg22 and 24-epi-brassinolide applications on FLS2 and BRI1 clusters, and included our new results in Figure 3 and Figure 4 in the revised manuscript. We believe these results improve our manuscript and we are thankful for the valuable comment.

Reviewer #1:

In my view, this is an important piece of work reporting a phenomenon suited to explain how common co-receptors serve surface receptors that are implicated in physiologically diverse programs, such as immunity and development. As this appears to be the first study ever to address such topic in higher organisms I clearly see the novelty and why it would make a valuable contribution to eLife. As I cannot really comment the quality and validity of the cell biological work I support publication of this work because of its eminent importance for the field and beyond.

We thank this reviewer for these very positive comments.

Reviewer #2:

FLS2 and BRI1 trigger different gene transcriptional profiles and physiological responses. However, they share common co-receptor SERKs. In addition, different members of RLCKs function differentially downstream of the RLK complexes. How the signaling specificity is governed at the RLK level is not clear. This work examined how FLS2 and BRI1 distribute on the PM mainly by imaging the GFP or mCherry-tagged proteins without ligand treatments. They concluded that FLS2 and BRI1 are in PM nanoclusters, majority of which are in a spatially not over-lapping manner at the steady state. FLS2, but not BRI1, co-localizes with specific remorin protein-markers for certain nanodomains. FLS2 and BRI1 are associated with the cytoskeleton with FLS2 linked to actin-myosin and BRI1 linked to cortical microtubules.

This work is interesting and provides some insights into the FLS2 and BRI1 signaling.

We thank this reviewer for these positive comments.

It is known that upon the ligand perception, FLS2 and BRI1 complex with SERK3 and activate the signaling. However, the majority of work here was done with FLS2 without flg22 treatment and examines the steady state of FLS2. BRI1 localization was done in a similar fashion-without BR treatment. Unlike FLS2 signaling, the endogenous BR signaling is constantly on in the cells (although likely at a low level). It remains unknown how the steady-state FLS2 patterns are related to the ligand-activated clusters and how this change may affect the signaling specificity, the key question that this manuscript is attempting to address. Recent work reported a highly dynamic behavior of FLS2 and BRI1 in particular in response to the cognate ligand perception (Somssich et al., Real-time dynamics of peptide ligand-dependent receptor complex formation in planta. Science Signaling 2015; Wang et al., Spatiotemporal dynamics of the BRI1 receptor and its regulation by membrane microdomains in living Arabidopsis cells. Molecular Plant 2015). The conclusion that FLS2 and BRI1 are largely in spatially separated nanoclusters at the steady state was based on the comparison between FLS2 and BRI1 localization in the absence of ligands in WT plants. The BR signaling is on in the absence of exogenous BR treatment. What they have observed in this experimental setting may contain both steady-state and activated form of BRI1. Additional experiments with BR biosynthetic mutants will be helpful to confirm the conclusion. Comparison the RLK patterns with and without ligands will provide insights into how these patterns contribute the signaling specificity.

We thank this reviewer for this valuable comment, and agree that some aspects of our manuscript require the absence or presence of ligands to allow full comparison between the FLS2 and BRI1 receptor populations. In order to compare both receptors in an inactive configuration, we now included BRZ treatments for the characterization of FLS2 and BRI1 clusters (new Figure 3), and also for the co-localisation studies between the two receptors (new Figure 5). Similarly, we studied FLS2 and BRI1 cluster behaviour after exogenous application of 24-epi-brassinolide or flg22, respectively (new Figure 3).

Considering that the nanodomains are usually 10-100 nm in size, the resolutions of current confocal microscopic techniques to differentiate the two objectives are rather limited (CLSM 500~700 nm). Cautious should be taken into account for the spatial separations of clusters.

We agree that the presented values for FLS2 and BRI1 cluster sizes are most likely overestimations as stated in our discussion. It can be expected that many of the observed receptor clusters are actually below the diffraction limit of roughly 250-300 nm for CLSM and VAEM. To determine the true size of FLS2 and BRI1 clusters, the application of super-resolution methods would be required. Currently, we are not able to apply these methods on our samples. We tried to investigate our samples using SIM (structured illumination microscopy), but unfortunately, it was technically not feasible.

With regard to the spatial separation of clusters, our inability to differentiate between objects closer than 250-300 nm means that we would actually underestimate the separation of small and nearby clusters. Therefore, this should not interfere with our current conclusions. Though, we hope to be able to revisit these questions with improved technical capabilities in the future.

As only 5~10% BRI1 are actively involved in BR signaling (van Esse et al., 2012), the BRI1 patterns are based on the observations from the "majority" of BRI1. At the moment, the knowledge is lacking on where those 5~10% activated portions are. The highly dynamic nature of BRI1 and FLS2 clusters upon the ligand perception will add another layer of complications.

Unfortunately, we are currently not able to address this important question, which receptors or receptor complexes among the total population are signalling active – neither for BRI1 nor for FLS2. Ligand perception most likely influences the dynamic behaviour of individual receptors or receptor clusters (see Figure 3 and Figure 4 of the revised manuscript). However, at the moment the outcomes are still controversial (as discussed in our discussion) and need to be specifically addressed in more details in future studies.

Regarding ligand perception by FLS2, the report of Somssich et al., 2015 showed that flg22 triggers first the association of FLS2 with BAK1 followed by the aggregation of FLS2/BAK1 complexes. Here, it is important to mention that the FRET approaches applied by Somssich et al., 2015 have a very high spatial resolution in the range of around 10 nm. In contrast, our confocal or VAEM techniques are diffraction-limited approaches, and thus provide only a spatial resolution of around 250-300 nm. Consequently, the temporal dynamics described by Somssich et al., 2015 do not necessarily require high spatial dynamics or long-distance molecular movements within the plasma membrane – they can occur within a single nanodomain harbouring both FLS2 and BAK1 molecules. Such a scenario is in line with our new results of increased receptor confinement after ligand application (new Figure 3 and Figure 4 of the revised manuscript). Additionally, these findings are in agreement with a previous study of FLS2 by Ali et al., 2007 and with findings for EGFR (Low-Nam et al., 2012), which both show that receptor mobility decreases upon ligand perception.

Regarding ligand perception by BRI1, Wang et al., 2015 indeed reported increased spatial receptor dynamics one hour after BL application in Arabidopsis roots. These results are in stark contrast to our observations and to findings for FLS2 and EGFR stated above. The reference has been included and discussed in our manuscript. Possible explanations for the apparent discrepancy between the results may be of technical and experimental nature. We limited our observation of receptor clusters to 30 min only (to avoid capturing FLS2 endocytosis, which most likely represents a post-ligand degradation step of the activated receptor), applied different imaging and data analysis settings, and studied the receptors in epidermal leaf cells.

Taken together, the well-established associations between FLS2, BRI1 and their co-receptors after ligand perception, as well as the findings reported by Somssich et al., 2015 and Ali et al., 2007, are in line with our observations of (at least initially) increased receptor confinement upon ligand binding.

Reviewer #3:

Bücherl et al. report that the immune receptor FLS2 and the brassinosteroid receptor BRI1 are found in distinct nanodomains of the plasma membrane (PM) that display differences in mobility and association to the cytoskeleton and contain different remorins. The topic is exciting and timely as understanding the spatiotemporal dynamics of receptor interactions will be key to understanding how specificity is generated at the PM.

We thank this reviewer for these positive comments, and for acknowledging the timeliness and importance of our study and its results.

1) Based on the similarity in cluster size observed in stable transgenic Arabidopsis line and after transient expression in N. benthamiana all following experiments are performed in N. benthamiana epidermal cells. In remains unclear if the two signalling pathways are operational and if endogenous receptors could interfere with the analysis.

As this comment concerns the use of the transient expression system, we refer this reviewer to our response to the second comment from the general editorial comments above. In particular, considering whether the two signalling pathways are operational, we refer to the recent publications of Charparro-Garcia, 2015, Bozkurt et al., 2015; Somssich et al., 2015; and Mbengue et al., 2016.

Regarding the presence of endogenous receptors on the analysis, we do not think that a withdrawal of endogenous populations would change the conclusions of our observations. Preliminary data indicate no major differences for FLS2-GFP expressed under the FLS2 promoter in either wild-type Col-0 or *fls2* mutant background (see Figure 11). Though, the presence of endogenous receptors may lead to reduced fluorescence in certain clusters, which would not affect the present conclusions.

Author response image 2.FLS2-GFP forms receptor clusters in *fls2* mutant background.The images show the plasma membrane localisation of pFLS2::FLS2-GFP stably transformed in the *fls2* mutant or Columbia wild type background, respectively. The images were acquired 5 days post germination using CLSM and the scale bars represents a distance of 5 µm.**DOI:**
http://dx.doi.org/10.7554/eLife.25114.024

2) The Pearson correlation coefficients for FLS2-GFP and FLS2-mCherry are moderate at best. Jarsch et al. report values of around 0,7 for individual remorins tagged with GFP and mRFP and values below 0,5 as observed for FLS2-GFP and FLS2-mCherry would be sitting on the fence of random distribution and positive correlation according to Jarsch et al. This could either be a technical issue regarding alignment of the CLSM that can easily be controlled by using multicolour fluorescent beads.

We agree with this reviewer that our absolute correlation values are moderate. The discrepancy to the reported values of Jarsch et al., 2014 stems on the one hand from the use of slightly different correlation factors. Jarsch et al., 2014 reported so-called ‘Overlap coefficients’ whereas we use ‘Pearson correlation coefficients’. Overlap coefficients range from 0 to 1 and random or no co-localisation is therefore still represented by positive values – in contrast to Pearson correlation that ranges from -1 to +1. To avoid confusion, we removed the misleading reference from this section in the revised manuscript.

On the other hand, we investigated different protein populations with different dynamic features. Remorin proteins seem to localise very stably to certain plasma membrane domains – in the range of minutes and above. The investigated FLS2 and BRI1 clusters in our manuscript, however, are in comparison very short-lived. The high temporal dynamics of these receptor clusters reduces the probability to find the two differently tagged receptor populations at the same time at the same spot. Additionally, the number of individual receptors per cluster seems to be rather low; thus, further decreasing the probability of spatio-temporal co-occupancy of specific PM nanodomains.

Although our Pearson correlation values are low to moderate in absolute numbers, they are still a valuable measure for our comparative approach. For this purpose, we also switched the reference from FLS2-mCherry to BRI1-mRFP for control experiments showing the same outcome.

Alternatively the low correlation coefficient could indicate that one of the two fusion proteins is non-functional and thus behaves differently. Has functionality of the FLS2-fusions been fully evaluated by mutant complementation? Given that the authors excluded the co-receptor BAK1 form their analysis as it has been shown that functionality of BAK1-GFP is compromised, the authors are obviously aware of the importance of this matter but I could not find convincing evidence for full functionality of FL2-XFP in the published literature.

With regard to the functionality or usability of the two FLS2 variants, we would like to refer to the recent publication of Mbengue et al., PNAS 2016.

3) To go beyond the level of high-resolution descriptive the authors would need to include data supporting that the observed nanoclusters are functionally relevant. What happens to signalling in remorin mutants? Are nanoclusters dependent on the cytoskeleton? How is the co-receptor BAK1 distributed and does absence of BAK1 affect nanocluster formation?

We absolutely agree with this reviewer that the next step is to characterise the functional importance of PM nanodomains. However, this is a whole project by itself, and goes beyond the scope of the current manuscript, whose main aim was to assess the comparative localization patterns of BRI1 (a steroid hormone receptor) and FLS2 (an immune receptor), which Reviewer 1 has already highlighted as being very important and timely.

The physiological characterisation of *rem* mutants is ongoing. It is however worthwhile to note that no significant phenotype has ever been reported so far for any single *rem* mutant in Arabidopsis, and thus that the generation of multiple mutants may be required. Similarly, studies on the interplay between cytoskeleton and receptor clusters are in progress. Preliminary data indicate that cluster formation is not abolished upon actin depolymerisation, for example; however, the influence on cluster dynamics is still unclear (see Figure 12).

Author response image 3.Actin depolymerisation does not abolish FLS2 cluster formation.**DOI:**
http://dx.doi.org/10.7554/eLife.25114.025

The images show the plasma membrane localisation of pFLS2::FLS2-GFP and p35S::LifeAct- tagRFP after depolymerisation of the actin cytoskeleton by 3.5 h incubation in 30 µM LatriculinB. The images were acquired 14 days post germination using CLSM and the scale bars represents a distance of 5 µm.

</Figure 12 title/legend>

As mentioned earlier (see response to general editorial comments), our included preliminary data (Figure 10) indicate the co-localisation of BAK1 with BRI1 and FLS2 in certain PM nanodomains. As part of another submitted manuscript (Hutten et al., under revision with PLoS One; co-last authors: Johannes Hohlbein & Jan Willem Borst; co-auther: CA Bücherl), we show that absence of SERK1 and BAK1 does not abolish cluster formation for BRI1. Further, we show here preliminary data indicating a similar situation for FLS2 (see Figure 13).

Author response image 4.FLS2 still undergoes cluster formation in *serk* mutant backgrounds.**DOI:**
http://dx.doi.org/10.7554/eLife.25114.026

The images show the plasma membrane localisation of pFLS2::FLS2-GFP in *bak1-4, bak1- 5/bkk1-1*, and Columbia wild type background, respectively. The images were acquired 5 days post germination using CLSM and the scale bars represents a distance of 5 µm.

</Figure 13 title/legend>

We hope this reviewer and the Editor will agree that, although these are very important questions, they remain beyond the scope of the current manuscript.

[Editors' note: the author responses to the re-review follow.]

The manuscript has been improved but there are two remaining issues that need to be addressed before acceptance, as outlined below:

1) The Pearsson correlation coefficient for FLS2-GFP vs FLS2-mCherry is surprisingly low. You argue that this is due to the low number of receptors per cluster and the dynamic behaviour of the clusters. This is one possible explanation, yet in the absence of a positive control technical reasons cannot be excluded.

Although the Pearson R correlation coefficients may appear low, our conclusions of co-localisation vs. non-co-localisation are always based on statistically significant differences as illustrated in our graphs and explained in the figure legends.

Also, on a more technical side, in noisy data, Pearson R correlation coefficients of (+/-) 0.3-0.5 are generally considered to be moderate, rather than small [typically guidelines for this are taken from the work of Jacob Cohen, see chapter 3, Cohen, J. (1988) Statistical power analysis for the behavioral sciences 2nd ed.]. Pearson’s R is different from the typical Spearman’s R in this respect. The data we are looking at is inherently noisy because of low receptor numbers per cluster so a moderate correlation is a good indicator of co-localisation. It is possible to interpret Pearson’s R as the proportion of the variability explained by the variable under test, given the noisy nature of our data then we think that we have accounted for the larger fraction of the variation.

2) Regarding functionality of FLS2-GFP the authors refer to Mbengue et al., 2016 yet one does not find evidence in there that demonstrates a wt-level flg22 response in the complementation line used.

While it is correct that we cannot necessarily claim that the FLS2-GFP and FLS2-mCherry fusion proteins used in this study and previous publications are as active as wild-type untagged FLS2, the fact that the FLS2-GFP and FLS2-mCherry undergo flg22-induced endocytosis indicates that they represent biologically-active receptors capable of perceiving the flg22 ligand and to become activated (and then later endocytosed) in response to flg22 perception. Then, as eluded to above, the main conclusions made in our manuscript is that FLS2-GFP and FLS2-mCherry co-localise based on a positive Pearson R correlation coefficient, while the comparison between FLS2-mCherry and BRI1-GFP, for example, yields a Pearson R correlation coefficient close to zero, with the difference between these Pearson R correlation coefficients being statistically significant.